# An effector of the Irish potato famine pathogen antagonizes a host autophagy cargo receptor

Yasin F Dagdas[1†], Khaoula Belhaj[1†], Abbas Maqbool[2], Angela Chaparro-Garcia[1], Pooja Pandey[3], Benjamin Petre[1], Nadra Tabassum[3], Neftaly Cruz-Mireles[1], Richard K Hughes[2], Jan Sklenar[1], Joe Win[1], Frank Menke[1], Kim Findlay[4], Mark J Banfield[2], Sophien Kamoun[1]*, Tolga O Bozkurt[1,3]*

[1]The Sainsbury Laboratory, Norwich, United Kingdom; [2]Department of Biological Chemistry, John Innes Centre, Norwich, United Kingdom; [3]Department of Life Sciences, Imperial College London, London, United Kingdom; [4]Department of Cell and Developmental Biology, John Innes Centre, Norwich, United Kingdom

**Abstract** Plants use autophagy to safeguard against infectious diseases. However, how plant pathogens interfere with autophagy-related processes is unknown. Here, we show that PexRD54, an effector from the Irish potato famine pathogen *Phytophthora infestans*, binds host autophagy protein ATG8CL to stimulate autophagosome formation. PexRD54 depletes the autophagy cargo receptor Joka2 out of ATG8CL complexes and interferes with Joka2's positive effect on pathogen defense. Thus, a plant pathogen effector has evolved to antagonize a host autophagy cargo receptor to counteract host defenses.

*For correspondence: sophien. kamoun@tsl.ac.uk (SK); o.bozkurt@imperial.ac.uk (TOB)

†These authors contributed equally to this work

Competing interests: The authors declare that no competing interests exist.

## Introduction

Autophagy is conserved catabolic pathway that sequesters unwanted cytosolic components into newly formed double membrane vesicles, autophagosomes, to direct them to the cell's lytic compartment (*He and Klionsky, 2009*). The process plays a vital role in survival of the organism by improving cellular adaptation to environmental and stress conditions (*Shintani and Klionsky, 2004*). Autophagy provides building blocks and energy for elementary cellular processes by degrading dysfunctional or unnecessary cellular components during nutrient deprivation (*Shintani and Klionsky, 2004*). However, even though autophagy was initially thought to be a bulk degradation process activated during starvation, recent studies showed that it can act selectively, capturing specific substrates through specialized cargo receptors to respond to a variety of environmental and stress conditions (*Stolz et al., 2014*).

Autophagy is executed through coordinated action of more than 30 core proteins known as the ATG (autophagy-related) proteins (*Lamb et al., 2013*). Selective autophagy is regulated through specific interactions of autophagy cargo receptors and ATG8 proteins (*Stolz et al., 2014*). Autophagy cargo receptors carry a short sequence motif called ATG8-interaction motif (AIM) that binds lipidated ATG8 proteins anchored on autophagosomal membranes. Cargo receptors mediate recognition of a diverse set of cargo (*Stolz et al., 2014*). For instance, mammalian autophagy cargo receptors NDP52 and optineurin can recognize intracellular pathogenic bacteria and mediate their autophagic removal by sorting the captured bacteria inside the ATG8-coated autophagosomes (*Boyle and Randow, 2013*). Nevertheless, the precise molecular mechanisms of selective autophagy and the components that regulate it remain unknown (*Huang and Brumell, 2014*; *Mostowy, 2013*; *Randow, 2011*).

**eLife digest** Plants and other living organisms can survive stress and starvation by digesting and recycling parts of their own cells. This process is known as autophagy and it involves engulfing cellular material inside spherical structures called autophagosomes, before delivering it to sites in the cell where digestive enzymes can break the material down. A form of autophagy, known as selective autophagy, can specifically degrade toxic substances such as disease-causing microbes. Selective autophagy works through proteins called autophagy cargo receptors that define which molecules are targeted for degradation. However, it was not clear whether autophagy protects plants from infections, or how much disease-causing microbes interfere with this process for their own benefit.

The microbe that causes late blight of potatoes (called *Phytophthora infestans*) is infamous for triggering widespread famines in Ireland in the 19th century. This disease-causing microbe continues to pose a serious threat to food security today, and parasitizes plant tissues by releasing proteins called effectors that enter the plant's cells to subvert the plant's physiology and counteract its defenses.

Dagdas, Belhaj et al. now report that an effector from *P. infestans,* called PexRD54, can bind to autophagy-related protein from potato, called ATG8CL, and stimulate the formation of autophagosomes. Further experiments revealed that the PexRD54 effector could outcompete a plant autophagy cargo receptor that would otherwise bind to ATG8CL. This plant cargo receptor contributes to the plant's defences, and by preventing it from interacting with ATG8CL, PexRD54 makes the plant more susceptible to infection by *P. infestans*.

These findings show that the PexRD54 effector has evolved to interact with an autophagy-related protein to counteract the plant's defences. Dagdas, Belhaj et al. suggest that PexRD54 might do this by activating autophagy to selectively eliminate some of the molecules that the plant use to defend itself. Furthermore, *P. infestans* might also benefit from the nutrients that are released when cellular material is broken down via autophagy. Future work could test these two hypotheses and explore whether other effectors from disease-causing microbes work in a similar way.

In plants, autophagy plays important roles in stress tolerance, senescence, development, and defense against invading pathogens (*Patel and Dinesh-Kumar, 2008*; *Lenz et al., 2011*; *Vanhee and Batoko, 2011*; *Li and Vierstra, 2012*; *Lv et al., 2014*; *Teh and Hofius, 2014*). Specifically, autophagy is implicated in the accumulation of defense hormones and the hypersensitive response, a form of plant cell death that prevents spread of microbial infection (*Yoshimoto et al., 2009*). However, the molecular mechanisms that mediate defense-related autophagy and the selective nature of this process are poorly understood. Furthermore, how adapted plant pathogens manipulate defense-related autophagy and/or subvert autophagy for nutrient uptake is unknown.

In this study, we investigated how a pathogen interferes with and coopts a plant autophagy pathway. The potato blight pathogen, *Phytophthora infestans*, is a serious threat to food security, causing crop losses that, if alleviated, could feed hundreds of millions of people (*Fisher et al., 2012*). This pathogen delivers RXLR-type effector proteins inside plant cells to enable parasitism (*Morgan and Kamoun, 2007*). RXLR effectors form a diverse family of modular proteins that alter a variety of host processes and therefore serve as useful probes to dissect key pathways for pathogen invasion (*Morgan and Kamoun, 2007*; *Bozkurt et al., 2012*). Here, we show that the RXLR effector PexRD54 has evolved to bind host autophagy protein ATG8CL to stimulate autophagosome formation. In addition, PexRD54 depletes the autophagy cargo receptor Joka2 out of ATG8CL complexes to counteract host defenses against *P. infestans*.

## Results and discussion

As part of an *in planta* screen for host interactors of RXLR effectors, we discovered that the *P. infestans* effector PexRD54 associates with ATG8CL, a member of the ATG8 family (Materials and methods, *Supplementary files 1,2*). The association between PexRD54 and ATG8CL was retained under stringent binding conditions in contrast to other candidate interactors (*Supplementary file 1*). We

validated the association with reverse coimmunoprecipitation after co-expressing the potato ATG8CL protein with the C-terminal effector domain of PexRD54 *in planta* (*Figure 1A,B*). In addition, PexRD54 expressed and purified from *Escherichia coli* directly bound ATG8CL in vitro with high affinity and in a one to one ratio ($K_D$ = 383 nM based on isothermal titration calorimetry) (*Figure 1C*). PexRD54 has two predicted ATG8 Interacting Motifs (AIMs) that match the consensus amino acid sequence W/F/Y-x-x-L/I/V (AIM1 and AIM2, *Figure 1A*). *In planta* coimmunoprecipitations of single and double AIM mutants of PexRD54 revealed that AIM2, which spans the last four amino acids of the protein (positions 378–381), is required for association with ATG8CL (*Figure 1D*). PexRD54^AIM2 mutant also failed to bind ATG8CL in vitro (*Figure 1E*). In addition, ATG8CL bound with high affinity to a synthetic peptide (KPLDFDWEIV) that matches the last 10 C-terminal amino acids of PexRD54 ($K_D$ = 220 nM) (*Figure 1—figure supplement 1*). We conclude that the C-terminal AIM of PexRD54 is necessary and sufficient to bind ATG8CL.

ATG8 occurs as a family of nine proteins in potato (*Figure 2—figure supplements 1–2*). PexRD54 bound ATG8CL with ~10 times higher affinity than another ATG8 family member, ATG8IL, in both *in planta* and in vitro assays (*Figure 2*). These findings prompted us to use ATG8IL as a negative control in the subsequent experiments.

We then investigated the subcellular localization of PexRD54 within plant cells. N-terminal fusions of PexRD54 to the green fluorescent protein (GFP) or red fluorescent protein (RFP) labelled the nucleo-cytoplasm and mobile punctate structures (*Figure 3—figure supplement 1* and *Video 1*). Immunogold labelling in transmission electron micrographs of cells expressing GFP:PexRD54 revealed a strong signal in electron dense structures that are not peroxisomes (*Figure 3—figure supplements 2–3*, *Dagdas et al., 2016*). To determine whether these structures are ATG8CL autophagosomes, we transiently co-expressed RFP:PexRD54 with GFP:ATG8CL in plant cells and observed an overlap between the two fluorescent signals in sharp contrast to RFP:PexRD54^AIM2 and RFP:EV negative controls (*Figure 3* and *Video 2*). This indicates that PexRD54 localizes to ATG8CL-marked autophagosomes and its C-terminal AIM is necessary for autophagosome localization. In contrast, RFP:PexRD54 signal overlapped with GFP:ATG8IL-labelled autophagosomes in only 15–20% of observations consistent with its weaker binding affinity to ATG8IL (*Figure 3—figure supplement 4*).

To further confirm that PexRD54-labelled endomembrane compartments are indeed autophagosomes, we investigated the effect of the autophagy inhibitor 3-methyl adenine (3-MA) (*Hanamata et al., 2013*) on PexRD54 localization. Compared to water, 3-MA treatment reduced the number of PexRD54 and ATG8CL puncta but did not reduce the number of puncta of the trans-Golgi network (TGN) marker VTI12 (*Geldner et al., 2009*) (*Figure 3—figure supplement 5*).

Phospholipid modification of a conserved glycine residue at the C-terminus of ATG8 proteins is required for autophagosome formation, and deletion of this terminal glycine yields a dominant negative ATG8 (*Hanamata et al., 2013*). We deployed a terminal glycine deletion mutant of ATG8CL (ATG8CLΔ) to determine its effect on subcellular distribution of PexRD54 (*Figure 3—figure supplement 6*). As expected, deletion of the terminal glycine did not affect binding of ATG8CL to PexRD54 (*Figure 3—figure supplement 7A*). However, GFP:ATG8CLΔ led to the depletion of RFP:PexRD54 labelled puncta presumably because the dominant negative effect of ATG8CLΔ prevented accumulation of RFP:PexRD54 in ATG8CL-labelled autophagosomes (*Figure 3—figure supplement 7B–D*). In contrast, GFP:ATG8ILΔ, a terminal glycine deletion mutant of ATG8IL, had no effect on the punctate localization of RFP:PexRD54 (*Figure 3—figure supplement 7C–D*). These experiments independently support the finding that PexRD54 accumulates in ATG8CL autophagosomes.

Increase in ATG8 labelled puncta is widely used as a functional readout of autophagic activity (*Hanamata et al., 2013*; *Bassham, 2015*). In samples expressing PexRD54, we noticed a ~fivefold increase in the number of ATG8CL marked autophagosomes compared to control samples expressing PexRD54^AIM2 or empty vector control (*Figure 4*, *Video 3*). In contrast, PexRD54 did not alter the number of ATG8IL autophagosomes consistent with the weak binding noted between these two proteins (*Figure 4A*). This indicates that PexRD54 stimulates the formation of ATG8CL autophagosomes.

Next, we set out to determine the effect of PexRD54 on autophagic flux. Treatment of RFP:ATG8CL expressing leaves with the specific vacuolar ATPase inhibitor concanamycin-A (*Bassham, 2015*) increased the number of ATG8CL-labelled puncta both in the presence of PexRD54 or controls (PexRD54^AIM2 or vector control) indicating that PexRD54 does not block autophagic flux

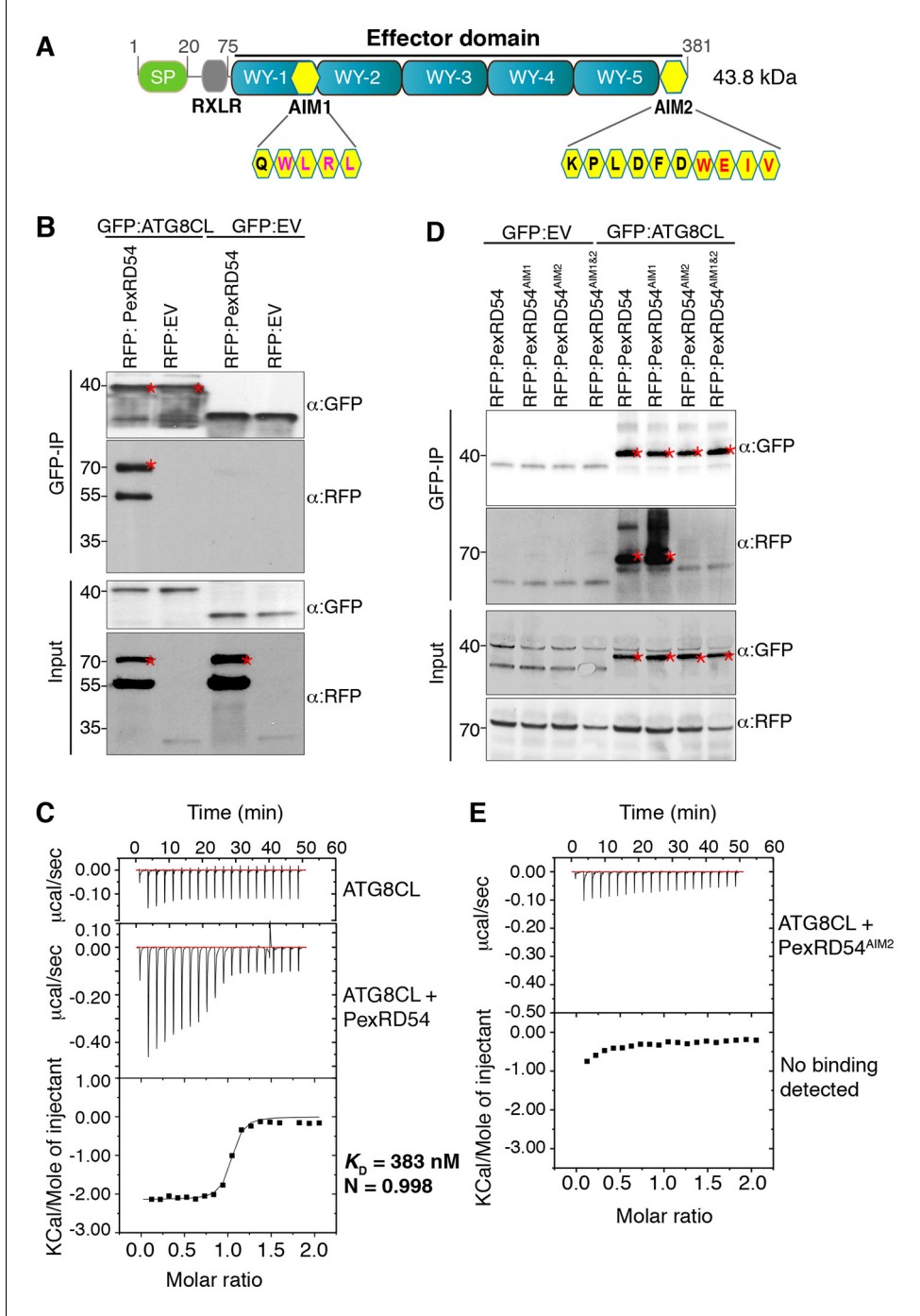

**Figure 1.** PexRD54 binds to ATG8CL via a C-terminal ATG8 interacting motif (AIM). (**A**) Domain organization of PexRD54. PexRD54 is a canonical RXLR effector with five WY folds (**Win et al., 2012**). The amino acid sequences of candidate AIMs are highlighted in yellow color and indicated in brackets. (**B**) Validation of PexRD54-ATG8CL association *in planta*. RFP:PexRD54 or RFP:EV (Empty vector) were transiently co-expressed with GFP:ATG8CL or GFP:EV in *N. benthamiana* leaves. Immunoprecipitates (IPs) obtained with anti-GFP antiserum and total protein extracts were immunoblotted with appropriate antisera. Stars indicate the expected band sizes. (**C**) PexRD54 binds ATG8CL in vitro. The binding affinity of PexRD54 to ATG8CL was determined using isothermal titration calorimetry (ITC). Upper panel shows heat differences upon injection of ATG8CL into buffer or PexRD54 and the bottom panel show integrated heats of injection (■) and the best fit (solid line) to a single site binding model using MicroCal Origin. $K_D$=383 nM, N=0.998, $\Delta H$= −8.966 kJ.mol$^{-1}$ and $\Delta S$ = 0.092 J.mol$^{-1}$.K$^{-1}$. The values of $K_D$, N, $\Delta H$ and $\Delta S$ are representative of two independent ITC experiments. (**D**) ATG8 Interacting Motif 2 (AIM2) mediates ATG8CL

*Figure 1 continued on next page*

*Figure 1 continued*

binding *in planta*. RFP:PexRD54, RFP:PexRD54[AIM1], RFP:PexRD54[AIM2] or RFP:PexRD54[AIM1&AIM2] were transiently co-expressed with GFP:ATG8CL or GFP:EV in *N. benthamiana* leaves. IPs obtained with anti-GFP antiserum and total protein extracts were immunoblotted with appropriate antisera. Stars indicate the expected band sizes.
(**E**) AIM2 mediates ATG8CL binding in vitro. The binding affinity of PexRD54[AIM2] to ATG8CL was determined using ITC. Upper panel shows heat differences upon injection of PexRD54[AIM2] and the bottom panel show integrated heats of injection (■). No binding was detected between PexRD54[AIM2] and ATG8CL.

The following figure supplement is available for figure 1:

**Figure supplement 1.** A synthetic peptide composed of the last ten amino acids of PexRD54 is sufficient for binding to ATG8CL.

(*Figure 5A–B*). We also confirmed these observations using western blot analyses. PexRD54, but neither PexRD54[AIM2] or vector control, increased the levels of GFP:ATG8CL protein 3 days after co-expression *in planta* (*Figure 5C*). Treatment of three-day samples with E64d, an inhibitor of vacuolar cysteine proteases (*Bassham, 2015*), further increased protein levels of GFP:ATG8CL. This further confirms that PexRD54 stimulates autophagy rather than blocking autophagic flux (*Figure 5C*). PexRD54 did not alter the accumulation of GFP:ATG8IL or control GFP protein, confirming that PexRD54 increases ATG8CL protein accumulation specifically (*Figure 5C*). Consistent with these observations, we noted an increase in GFP:ATG8CL levels, but not in control GFP, during *P. infestans* infection relative to the mock infection (*Figure 5—figure supplement 1*).

The presence of a functional AIM in PexRD54 prompted us to hypothesize that this effector perturbs the autophagy cargo receptors of its host plants. Recently, Joka2 was reported as a selective autophagy cargo receptor of Solanaceous plants that also binds ATG8 via an AIM (*Svenning et al., 2011*; *Zientara-Rytter et al., 2011*) (*Figure 6A*). Indeed, *in planta* coimmunoprecipitation assays confirmed that potato Joka2, but not the AIM mutant, Joka2[AIM], associated with ATG8CL (*Figure 6—figure supplement 1*). Joka2 association with ATG8CL was somewhat specific given that this cargo receptor failed to coimmunoprecipitate with ATG8IL (*Figure 6—figure supplement 2*). Joka2, but not Joka2[AIM], also markedly increased the number of GFP:ATG8CL autophagosomes (*Figure 6—figure supplement 3*, *Video 4*), and enhanced ATG8CL protein levels (*Figure 6—figure supplement 4*). This indicates that Joka2 also activates ATG8CL-mediated selective autophagy.

Given that both PexRD54 and Joka2 bind ATG8CL via their respective AIMs, we hypothesized that PexRD54 interferes with the Joka2-ATG8CL complex. We tested our hypothesis by performing coimmunoprecipitation experiments between Joka2:RFP and GFP:ATG8CL in the presence or absence of PexRD54. Remarkably, ATG8CL complexes were depleted in Joka2 in the presence of PexRD54 relative to the PexRD54[AIM2] and vector control (*Figure 6—figure supplement 5*). Consistently, Joka2 binding to ATG8CL decreased with increasing PexRD54 concentrations (*Figure 6B*). The distinct AIMs of PexRD54 and Joka2 presumably determine the effect observed in these competition experiments. To further test this, we replaced the functional PexRD54 AIM with two sequences that cover the Joka2 AIM:GVAEWDPI (PexRD54[J2AIM1]) and GVAEWDPILEELKEMG (PexRD54[J2AIM2]) (*Figure 6C*). Both PexRD54[J2AIM1] and PexRD54[J2AIM2] associated with ATG8CL to a lesser extent than wild-type PexRD54 (*Figure 6D*), and were less effective than PexRD54 in depleting Joka2 out of ATG8CL complexes (*Figure 6—figure supplement 6*). These findings reveal that PexRD54 antagonizes Joka2 for ATG8CL binding.

Finally, we investigated the degree to which activation of Joka2-ATG8CL-mediated autophagy contributes to pathogen defense. Overexpression of Joka2, but not Joka2[AIM], significantly restricted the size of the disease lesions caused by *P. infestans* (*Figure 7A–B*). Conversely, virus-induced gene silencing of Joka2 resulted in increased disease lesions (*Figure 7—figure supplement 1*). This indicates that Joka2-mediated selective autophagy contributes to defense against this pathogen. Remarkably, PexRD54 counteracted the enhanced resistance conferred by Joka2 whereas PexRD54[AIM2] failed to reverse this effect (*Figure 7C–D*). We conclude that PexRD54 counteracts the positive role of Joka2-mediated selective autophagy in pathogen defense.

As demonstrated in mammalian systems, eukaryotic cells employ autophagy to defend against invading pathogens (*Boyle and Randow, 2013*; *Randow and Youle, 2014*). In turn, pathogens can

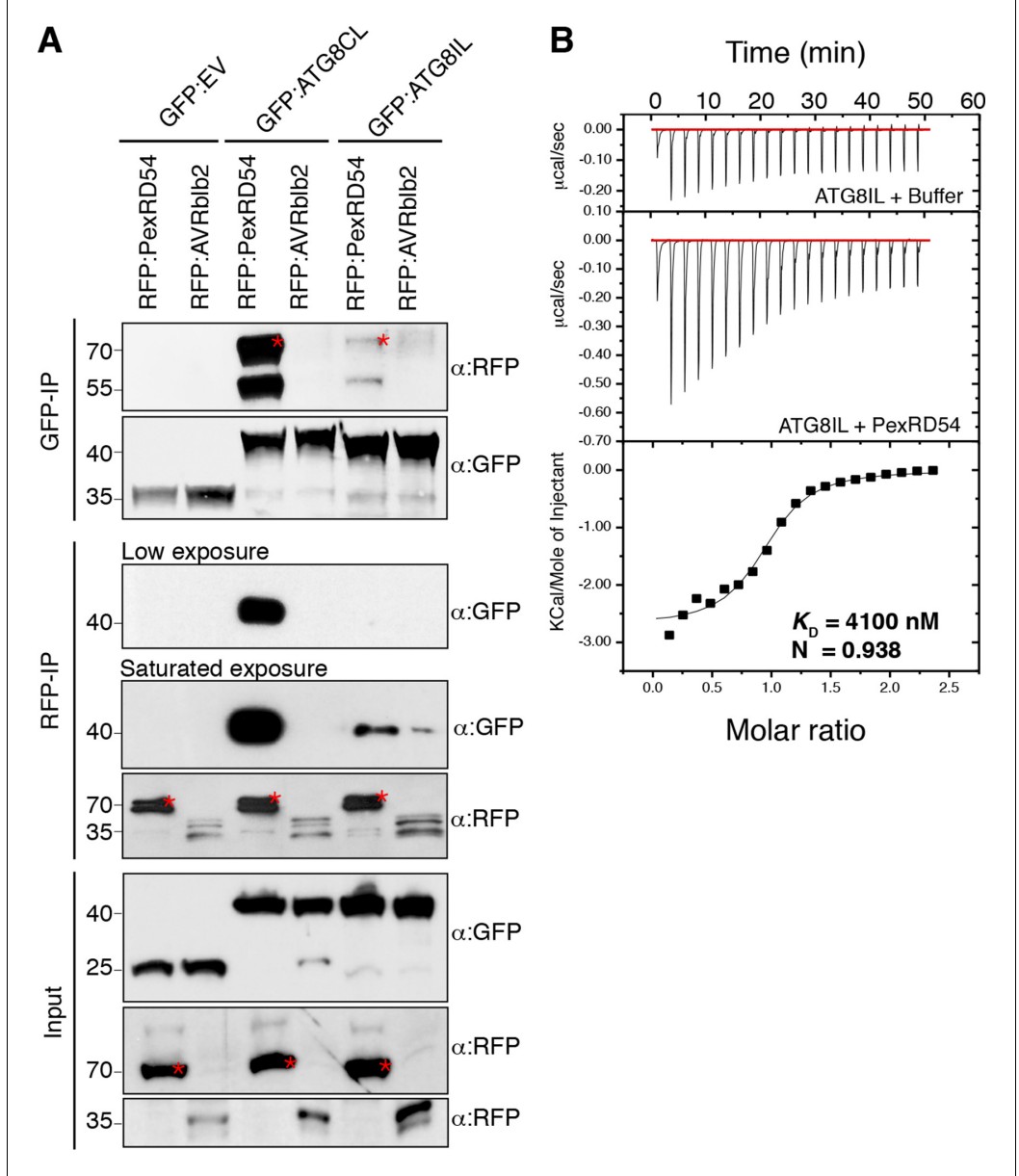

**Figure 2.** PexRD54 has higher binding affinity to ATG8CL than ATG8IL. (**A**) RFP:PexRD54, RFP:AVRblb2 or RFP:EV were transiently co-expressed with GFP:ATG8CL, GFP:ATG8IL or GFP:EV in *N. benthamiana* leaves and proteins were extracted two days after infiltration and used in immunoprecipitation experiments (IPs). IPs obtained with anti-GFP or anti-RFP antisera and total protein extracts were immunoblotted with appropriate antisera. RFP: AVRblb2 (*Bozkurt et al., 2011*), an RFP fusion to a different *P. infestans* RXLR effector, did not associate with ATG8CL or ATG8IL. Both the GFP and RFP IPs indicate higher binding affinity of PexRD54 to ATG8CL than ATG8IL. Stars indicate the expected band sizes. (**B**) PexRD54 has lower binding affinity to ATG8IL in vitro. The binding affinity of PexRD54 to ATG8IL was determined using isothermal titration calorimetry (ITC). Upper panel shows heat generated upon injection of ATG8IL into buffer or PexRD54 and lower panel shows integrated heats of injection (■) and the best fit (solid line) to a single site binding model using MicroCal Origin. The values of $K_D$ = 4100 nM, N = 0.938, $\Delta$H = −11.305 kJ.mol$^{-1}$ and $\Delta$S= 0.064 J.mol$^{-1}$K$^{-1}$ are representative values of two independent ITC experiments. The data show that PexRD54 binds ATG8CL with ~10 times higher affinity than ATGIL.

The following figure supplements are available for figure 2:

**Figure supplement 1.** Amino acid sequence alignment of ATG8 proteins from *Arabidopsis thaliana* (At), *Solanum tuberosum* (St), *Solanum lycopersicum* (Sl) and *Nicotiana benthamiana* (Nb).

**Figure supplement 2.** Phylogenetic analysis of ATG8 proteins.

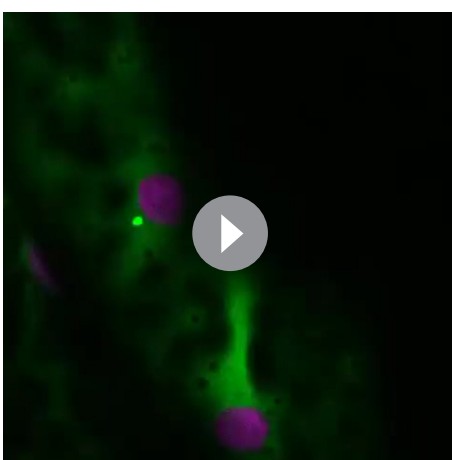

**Video 1.** PexRD54 localizes to mobile endomembrane compartments. GFP:PexRD54 was transiently expressed in *N. benthamiana* leaves and examined by confocal laser scanning microscopy, 3 days post infiltration. (Related to *Figure 3*)

deploy effectors to avoid autophagy and enable parasitic infection (*Baxt et al., 2013*). For instance, to counteract antimicrobial autophagy, intracellular bacterial pathogen *Legionella pneumophila* secretes a type IV effector protein RavZ that impedes autophagy by uncoupling ATG8-lipid linkage (*Choy et al., 2012*). In this study, we show that a plant pathogen effector has evolved an ATG8 interacting motif to bind with high affinity to the autophagy protein ATG8CL and stimulate the formation of ATG8CL-marked autophagosomes. Unlike the Legionella effector RavZ, PexRD54 activates selective autophagy possibly to eliminate defense-related compounds or to reattribute cellular resources by promoting nutrient recycling. Our results show that, in addition to disrupting, pathogens can also activate autophagy for their own benefit (*Figure 8*).

Additionally, we show that the effector competes with the host cargo receptor Joka2 and depletes it out of ATG8CL autophagosomes to promote disease susceptibility. Thus *P. infestans* coopts the host cell's endomembrane compartment to promote its own growth at the cost of the cell's physiology (*Bozkurt et al., 2015*). Joka2 could contribute to immunity by inducing autophagic removal of plant or pathogen molecules that negatively affect host defenses. It will be interesting in the future to determine the identity of potential defense-related cargo carried by Joka2 (*Figure 8*).

The physiological roles of selective autophagy and the molecular mechanisms involved remain to be determined both in plants and animals. Characterization of additional host cargo receptors and interactome analysis of particular ATG8 proteins should improve our understanding of how selective autophagy operates in response to a variety of stress conditions including immunity. In the potato genome, we identified nine ATG8 genes. Both Joka2 and the effector showed higher affinity to ATG8CL compared to ATG8IL (*Figure 2* and *Figure 6—figure supplement 2*). This indicates that variation between plant ATG8 proteins may contribute to the selective nature of autophagy.

This work further highlights the intricate changes in endomembrane compartment formation that take place during plant-microbe interactions (*Bozkurt et al., 2015*; *Lipka and Panstruga, 2005*; *Kwon et al., 2008*; *Ivanov et al., 2010*; *Wang et al., 2010*). Future studies will need to consider pathogen-directed modulation of the spatio-temporal dynamics of autophagy and subcellular trafficking during host infection. It will be interesting to determine whether other plant pathogens also secrete effectors that evolved an ATG8 interacting motif or target autophagy in other ways. These effectors would serve as valuable tools to dissect the mechanisms of defense-related selective autophagy.

## Materials and methods

### Identification of candidate ATG8 interacting motifs in PexRD54

ATG8 interacting motif (AIM), also known as LC3 interacting region (LIR), mediates interaction of cargo receptors or adaptors with ATG8 proteins anchored in autophagosome membranes (*Birgisdottir et al., 2013*). It follows the W/F/Y-xx-L/I/V amino acid consensus. Initially, we identified two AIM candidates in PexRD54, AIM1 and AIM2 based on the manual search of the consensus sequence mentioned earlier. Then, we confirmed these AIM candidates using the recently published iLIR software (*Kalvari et al., 2014*). This software assigned AIM1 with an iLIR score of 12 ($1.1e^{-1}$) and AIM2 with an iLIR score of 23 ($3.2e^{-3}$).

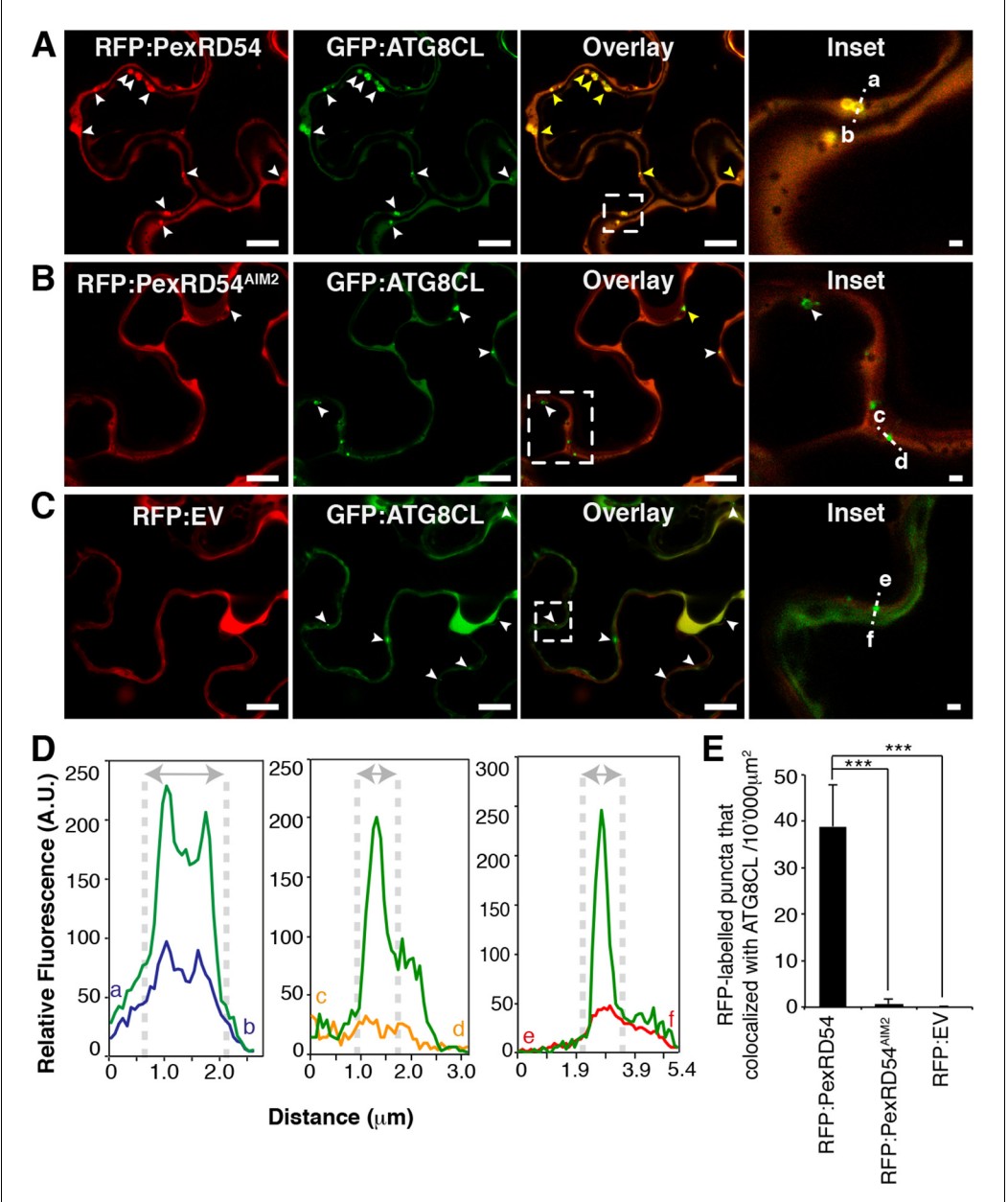

**Figure 3.** PexRD54 localizes to ATG8CL labelled autophagosomes. Transient co-expression of GFP:ATG8CL with (**A**) RFP:PexRD54, (**B**) RFP:PexRD54$^{AIM2}$ and (**C**) RFP control (RFP:EV) in *N. benthamiana* leaves. Confocal micrographs show single optical sections of RFP:PexRD54, RFP:PexRD54$^{AIM2}$ and RFP:EV in red, GFP:ATG8CL in green and the overlay indicating colocalization in yellow. White arrowheads point to punctate structures and yellow arrowheads point to puncta where GFP and RFP signals overlap. Far right panels highlight the dotted square region focusing on GFP:ATG8CL labelled puncta in overlaid GFP/RFP channels. Scale bar = 10 μm; scale bar in inset = 1 μm. (**D**) The intensity plots represent relative GFP and RFP fluorescence signals along the dotted line connecting points a-b, c-d and e-f that span GFP:ATG8CL marked puncta at far right panels. GFP: ATG8CL fluorescence intensity peak overlapped with fluorescence intensity peak of RFP:PexRD54 (left panel) but not with RFP:PexRD54$^{AIM2}$ (mid panel) or RFP:EV (right panel) validating the localization of RFP:PexRD54 at GFP:ATG8CL labelled autophagosomes. (**E**) Bar charts showing colocalization of GFP:ATG8CL puncta with RFP:PexRD54, RFP:PexRD54$^{AIM2}$ or RFP:EV punctate structures. Data are representative of 500 individual images from two biological replicates. Each replicate consists of five independent Z stacks with 50 images each, acquired from five independent leaf areas. (***p<0.001). Error bars represent ± SD.

The following figure supplements are available for figure 3:

**Figure supplement 1.** PexRD54 localizes to vesicle like structures.

*Figure 3 continued on next page*

*Figure 3 continued*

**Figure supplement 2.** PexRD54 localizes to high electron dense structures.

**Figure supplement 3.** PexRD54 labelled high electron dense structures that are not peroxisomes.

**Figure supplement 4.** PexRD54 labelled puncta rarely include GFP:ATGIL signal.

**Figure supplement 5.** Inhibition of autophagy hinders PexRD54 vesicular distribution.

**Figure supplement 6.** Conservation of terminal glycine residue in ATG8 proteins from various species.

**Figure supplement 7.** ATG8CL terminal glycine deletion mutant prevents formation of PexRD54 labelled puncta in a dominant negative manner.

## Sequence analyses and identification of ATG8CL

To determine the ATG8 variant(s) specifically targeted by PexRD54 among various host ATG8 family members, we performed a BLASTP search (*Altschul et al., 1990*) against solanaceous plant proteomes including *Solanum tuberosum* (potato), *Solanum lycopersicum* (tomato) and *Nicotiana benthamiana* using *Arabidopsis thaliana* ATG8C as the query protein sequence. We found nine ATG8 members in *S. tuberosum*, seven in *S. lycopersicum*, and eight in *N. benthamiana*. To verify the gene calls and open reading frame predictions of the putative orthologs in these three species, we performed a sequence alignment of the family members in each species using the Clustal X program (v2) (*Larkin et al., 2007*) and compared it to the published *A. thaliana* ATG8 sequences. We found evidence of misannotation for two *S. lycopersicum* sequences (Solyc10g006270 and Solyc08g078820) and three others in *N. benthamiana* (NbS00015425g0005, NbS00003316g0005 and NbS00003005g0010). The two *S. lycopersium* sequences and the NbS00003005g0010 sequence from *N. benthamiana* carried extra nucleotides before the likely start codon, which were corrected accordingly. A TBLASTN search against the *N. benthamiana* scaffolds followed by a BLASTP search of the different exons allowed curation of the two other sequences. Clustal X program was used for multiple sequence alignment of ATG8 variants. Boxshade server (http://embnet.vital-it.ch/software/BOX_form.html) was used to visualize the sequence alignment.

## Phylogenetic analyses

The phylogenetic tree of ATG8 homologs in plants was constructed with the neighbor-joining method using ATG8-like curated proteins from *S. tuberosum*, *S. lycopersium*, *N. benthamina*, and *A. thaliana*. The phylogenetic tree was constructed using MEGA5 (*Kumar et al., 2001*) with bootstrap values based on 1000 iterations. ATG8CL variants in *N. benthamiana* and potato have identical amino acid sequences.

## Cloning procedures and plasmid constructs

All primers used in this study are listed in *Supplementary file 3*. PexRD54 and Joka2 were amplified using polymerase chain reaction (PCR) from genomic DNA of *Phytophthora infestans* isolate T30-4 and *Solanum tuberosum* cv. Désirée cDNA, respectively. To amplify both amplicons, we used Phusion proof reading polymerase (New England Biolabs, UK) and primer pairs listed in *Supplementary file 3*. All amplicons were subsequently cloned into the pENTR/D-Topo Gateway

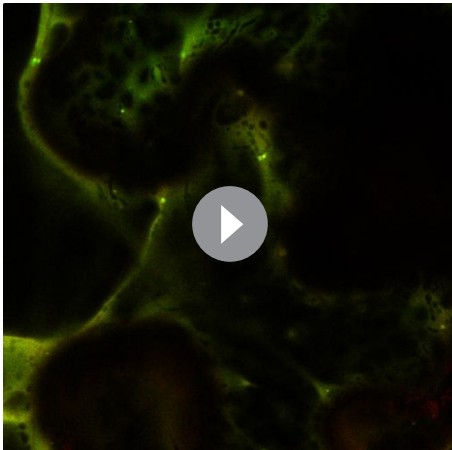

**Video 2.** PexRD54 and ATG8CL colocalize at mobile endomembrane compartments. RFP:PexRD54 was transiently co-expressed with GFP:ATG8CL in *N. benthamiana* leaves and examined by confocal laser scanning microscopy, 3 days post infiltration. (Related to *Figure 3*)

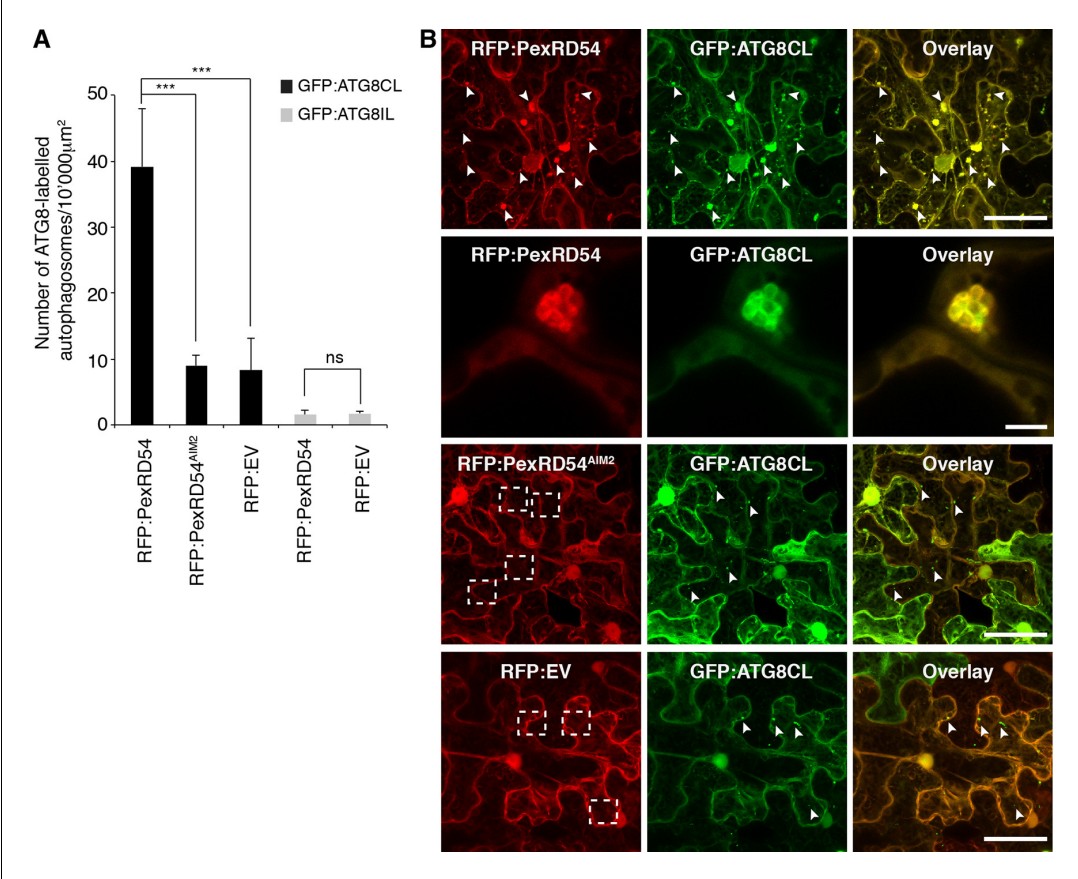

**Figure 4.** PexRD54 increases the number of ATG8CL autophagosomes. (**A**) Co-expression of RFP:PexRD54, but not RFP:PexRD54$^{AIM2}$ or RFP:EV significantly enhanced the number of GFP:ATG8CL-labelled autophagosomes in *N. benthamiana*. Bar charts display the number of GFP:ATG8CL or GFP:ATG8IL-labelled autophagosomes in the presence of RFP:PexRD54, RFP:PexRD54$^{AIM2}$ or RFP:EV. GFP:ATG8CL autophagosomes were significantly enhanced by the expression of RFP:PexRD54 (***p<0.001). GFP:ATG8IL autophagosomes were not significantly enhanced by expression of RFP: PexRD54 (ns=statistically not significant). The data are representative of 500 individual images from two biological replicates. Each replicate consists of five independent Z stacks with 50 images each, acquired from five independent leaf areas. (**B**) GFP:ATG8CL was transiently co-expressed with RFP: PexRD54, RFP:PexRD54$^{AIM2}$ or RFP:EV in *N. benthamiana* leaves and examined by confocal laser scanning microscopy 3 days post infiltration. Maximum projections of images show that RFP:PexRD54 increases the number of GFP:ATG8CL- labelled autophagosomes. RFP:PexRD54$^{AIM2}$ and RFP:EV were used as controls. Arrowheads point to punctate structures. Regions where GFP or RFP-labelled puncta do not overlap are indicated with dotted squares. Scale bar=50 µm. Zoomed single plane images shown in the second panel indicate that larger puncta co-labelled by RFP:PexRD54 and GFP: ATG8CL are ring-shaped autophagosome clusters. Scale bar=10 µm.

entry vector (Invitrogen, UK). The ATG8CL, ATG8CL*Δ*, ATG8IL, and ATG8IL*Δ* entry clones were custom synthesized into pUC57-Amp$^R$ using the sequences matching *S. tuberosum ATG8CL* or *ATG8IL* genes flanked by attL1 and attL2 gateway sites (Genewiz, UK). The destination constructs GFP: PexRD54, RFP:PexRD54, Joka2:RFP, Joka2:GFP, RFP:ATG8CL, GFP:ATG8CL, GFP:ATG8CL*Δ*, GFP: ATG8IL*Δ*, HA:PexRD54 were generated by Gateway LR recombination reaction (Invitrogen) of the corresponding entry clone and Gateway destination vectors pH7WGR2 (N-terminal RFP fusion), pK7WGF2 (N-terminal GFP fusion), pB7FWR2 (C-terminal GFP fusion), pB7RWG2 (C-terminal RFP fusion) and pK7WGF2 (N-terminal HA fusion, generated in house by replacement of GFP with an HA tag), respectively (*Karimi et al., 2002*). The GFP:PexRD54$^{AIM2}$, RFP:PexRD54$^{AIM1}$, RFP:PexRD54$^{AIM2}$, RFP:PexRD54$^{AIM1+2}$ RFP:PexRD54$^{KPLDFDWEIV}$, RFP:PexRD54$^{J2AIM1}$, RFP:PexRD54$^{J2AIM2}$, HA:PexR-D54$^{AIM1}$, HA:PexRD54$^{AIM2}$, HA:PexRD54$^{J2AIM1}$, HA:PexRD54$^{J2AIM2}$, Joka2$^{AIM}$:RFP constructs were generated as follows:PexRD54 and Joka2 mutant constructs (PexRD54$^{AIM1}$, PexRD54$^{AIM2}$, PexRD54$^{AIM1+2}$, PexRD54$^{KPLDFDWEIV}$, RFP:PexRD54$^{J2AIM1}$, RFP:PexRD54$^{J2AIM2}$, and Joka2$^{AIM}$:RFP) were cloned into the pENTR/D-Topo Gateway entry vector (Invitrogen) by PCR amplification with the primers carrying desired mutations (*Supplementary file 3*) using PexRD54 and Joka2 entry

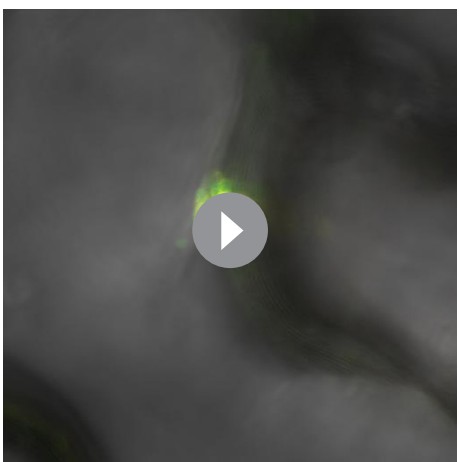

**Video 3.** RFP:PexRD54 and GFP:ATG8CL colocalize at mobile ring shaped clusters. RFP:PexRD54 was transiently co-expressed with GFP:ATG8CL in *N. benthamiana* leaves and examined by confocal laser scanning microscopy, 3 days post infiltration. (Related to *Figure 4*)

clones as templates followed by TOPO cloning procedure (Invitrogen). Templates were then eliminated by one-hour *Dpn*-I (New England Biolabs) restriction digestion at 37°C. Next, the entry clones of PexRD54 and Joka2 mutants were recombined into destination vectors pH7WGR2 or pB7RWG2 by Gateway LR reaction (Invitrogen). pTRBO FLAG:PexRD54 construct used for coimmunoprecipitations was custom synthesized (Genscript, , New Jersey, USA).

## Generation of ATG8 interacting motif (AIM) mutants

To generate ATG8 interacting motif (AIM) mutants in PexRD54 and Joka2, the conserved tryptophan and leucine residues of the canonical AIMs were both mutated to Alanine as previously done (*Zientara-Rytter et al., 2011*). In PexRD54[AIM1], PexRD54[AIM2] and Joka2[AIM], the AIM motifs e.g. 'WLRL', 'WEIV' and 'WDPI' were mutated to 'ALRA', 'AEIA' and 'ADPA', respectively.

## Biological material

*Nicotiana benthamiana* plants were grown and maintained throughout the experiments in a greenhouse at 22–25°C with high light intensity. *Phytophthora infestans* cultures were grown in plates with rye sucrose agar (RSA) media for 12–14 days as described elsewhere (*Song et al., 2009*). Sporangia were harvested from plates using cold water and zoospores were collected 1–3 hr after incubation at 4°C. Infection assays were performed by droplet inoculations of zoospore solutions of *P. infestans* on 3–4-week-old detached *N. benthamiana* leaves as described previously (*Song et al., 2009*; *Saunders et al., 2012*). For all infection assays, *P. infestans* isolate 88,069 was used.

### Transient gene-expression assays *in N. benthamiana*

Transient gene-expression *in planta* was performed by delivering T-DNA constructs with *Agrobacterium tumefaciens* GV3101 strain into 3–4-week-old *N. benthamiana* plants as described previously (*Bozkurt et al., 2011*). For transient co-expression assays, *A. tumefaciens* strains carrying the plant expression constructs were mixed in a 1:1 ratio in agroinfiltration medium [10 mM $MgCl_2$, 5 mM 2-(N-morpholine)-ethanesulfonic acid (MES), pH 5.6] to a final $OD_{600}$ of 0.2, unless otherwise stated.

### Coimmunoprecipitation (Co-IP) experiments and liquid chromatography tandem mass spectrometry (LC-MS/MS) analysis

Proteins were transiently expressed by *A. tumefaciens*-mediated transient expression (agroinfiltration) in *N. benthamiana* leaves and harvested 2 or 3 days post infiltration. Co-IP experiments and preparation of peptides for liquid chromatography–tandem mass spectrometry (LC-MS/MS) was performed as described previously (*Bozkurt et al., 2011*). Except the stringent PexRD54 immunoprecipitation assay (Stringent IP), all IPs were done using 150 mM NaCl buffer and 0.15% detergent concentration. For the stringent IP, the concentrations of salt and the detergent were increased to 250 mM and to 0.5%, respectively. LC-MS/MS analysis was performed with a LTQ Orbitrap mass spectrometer (Thermo Fisher Scientific, UK) and a nanoflow-HPLC system (nanoACQUITY; Waters Corp., UK) as described previously (*Petre et al., 2015*). LC-MS/MS data processing and protein identification were done as described previously (*Petre et al., 2015*).

*In planta* association of PexRD54 and Joka2 with either ATG8CL, ATG8CLΔ or ATG8IL constructs was tested by co-IP assays as follows: constructs were transiently co-expressed in *N. benthamiana* leaves by agroinfiltration followed by protein extraction 2–3 days post infiltrations. Protein extraction, purification and western blot analysis steps were performed as described previously

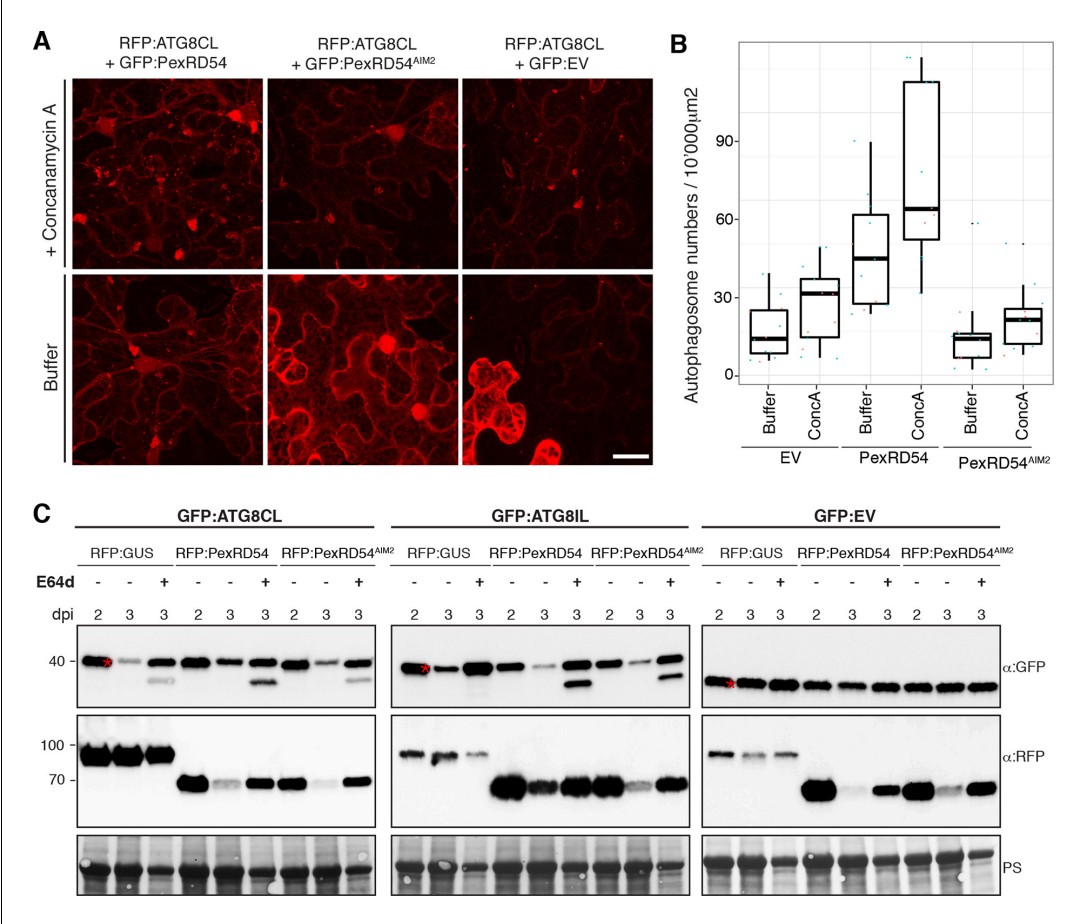

**Figure 5.** PexRD54 does not block autophagic flux. (A–B) ConcanamycinA treatment increases the number of autophagosomes in PexRD54 expressing samples.RFP:ATG8CL was transienly coexpressed with PexRD54, PexRD54$^{AIM2}$ and empty vector controls in *N. benthamiana*. Two days after infiltration, leaves were treated with concanamycinA (conA) or infiltration buffer and number of autophagosomes was counted 24 hr after treatment. ConA treatment significantly increased the number of autophagosomes in PexRD54 expressing cells (p<0.05), confirming PexRD54 does not block autophagic flux. Scale bar=10 μm. (C) E64D treatment increases ATG8CL protein levels in PexRD54 expressing samples. GFP:ATG8CL, GFP:ATG8IL and GFP:EV were transiently coexpressed with RFP:GUS, RFP:PexRD54 or RFP:PexRD54$^{AIM2}$ in *N. benthamiana* leaves and protein levels in total extracts were determined two and 3 days post infiltration (dpi). RFP:PexRD54 increased protein levels of GFP:ATG8CL but not GFP:ATG8IL consistent with stronger binding affinity of PexRD54 to ATG8CL. RFP:PexRD54 did not increase protein levels of GFP:EV, suggesting that protein level increase depends on ATG8CL binding and that PexRD54 does not increase protein levels in general. The samples were also treated with E64d to measure autophagic flux. In RFP:PexRD54 coexpressed 3 dpi samples, E64d treatment increased ATG8CL protein levels even more suggesting PexRD54 does not block autophagic flux. Hence, protein level increase is a result of stimulation of autophagy. The blots were stained with Ponceau stain (PS) to show equal loading.

The following figure supplement is available for figure 5:

**Figure supplement 1.** GFP:ATG8CL protein levels increase *in N. benthamiana* during *P. infestans* infection.

(*Saunders et al., 2012*; *Bozkurt et al., 2011*; *Oh et al., 2009*). Monoclonal FLAG M2 antibody (Sigma-Aldrich, UK), polyclonal GFP/RFP antibodies (Invitrogen, UK), and polyclonal HA (Sigma-Aldrich, UK) antibody were used as primary antibodies, and anti-mouse antibody (Sigma-Aldrich, UK) and anti-rat (Sigma-Aldrich, UK) antibodies were used as secondary antibodies.

## Determination of ATG8CL protein levels

To test if PexRD54 increased protein levels of ATG8CL, GFP:ATG8CL was co-expressed with RFP: GUS, RFP:PexRD54 and RFP:PexRD54$^{AIM2}$ in *N. benthamiana* leaves. Leaf samples were collected 2 and 3 days after infiltration. Total proteins were extracted as described previously (*Saunders et al., 2012*; *Oh et al., 2009*) and immunoblots were performed using the appropriate antisera. The same

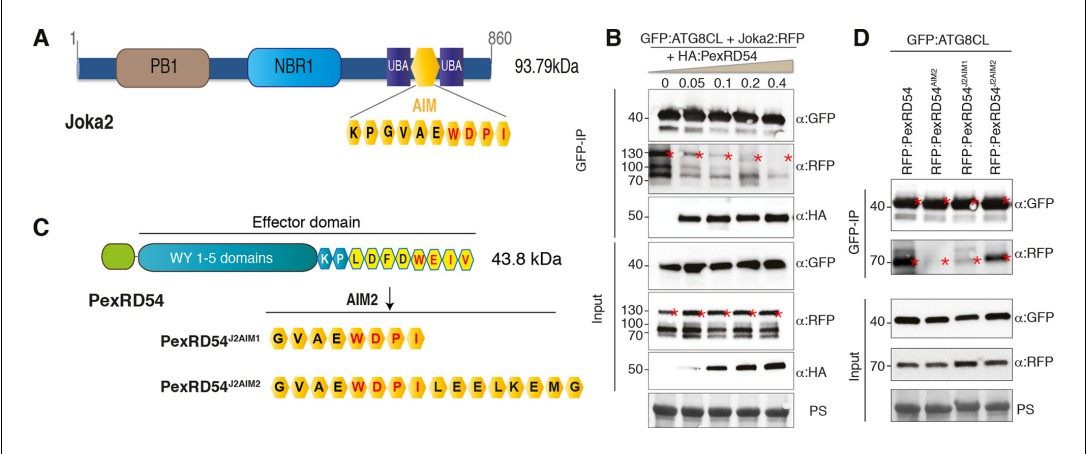

**Figure 6.** PexRD54 is a competitive antagonist of the plant selective autophagy cargo receptor Joka2. (**A**) Domain organization of Joka2. (**B**) PexRD54 reduces binding of Joka2 to ATG8CL in a dose-dependent manner. GFP:ATG8CL (OD=0.2) and Joka2:RFP (OD=0.2) were transiently co-expressed with varying Agrobacterium concentrations (from OD=0 to OD=0.4) carrying HA:PexRD54 construct in *N. benthamiana* (OD=optical density of Agrobacterium cells). Joka2:RFP is depleted in GFP:ATG8CL pulldowns as the expression of HA:PexRD54 increased. Immunoprecipitates (IPs) obtained with anti-GFP antiserum and total protein extracts were immunoblotted with appropriate antisera. Stars indicate expected band sizes. (**C**) Schematic illustrations of PexRD54 variants (PexRD54[J2AIM1] and PexRD54[J2AIM2]) with Joka2 AIM peptides. (**D**) Replacement of PexRD54 AIM with Joka2 AIM fragments decreases ATG8CL binding affinity. Immunoblots showing binding affinity of PexRD54, PexRD54[AIM2], PexRD54[J2AIM1] or PexRD54[J2AIM2] to GFP:ATG8CL. IPs obtained with anti-GFP antiserum and total protein extracts were immunoblotted with appropriate antisera. Stars indicate expected band sizes.

The following figure supplements are available for figure 6:

**Figure supplement 1.** Joka2 has a functional ATG8 interacting motif (AIM).

**Figure supplement 2.** Joka2 has higher binding affinity to ATG8CL than ATG8IL.

**Figure supplement 3.** Joka2 increases the number of ATG8CL labelled autophagasomes.

**Figure supplement 4.** Joka2 increases ATG8CL protein levels.

**Figure supplement 5.** PexRD54 outcompetes Joka2 for ATG8CL binding.

**Figure supplement 6.** PexRD54 AIM has a higher affinity to ATG8CL than Joka2 AIM.

experimental setup was also used to test whether Joka2 increases protein levels of ATG8CL. Joka2:RFP, Joka2[AIM]:RFP and RFP:GUS constructs were co-expressed with GFP:ATG8CL construct in *N. benthamina* leaves and total proteins were extracted 2 and 3 days after infiltration. Immunoblots were developed using appropriate antisera. To assay ATG8CL protein levels during infection with *P. infestans*, *N. benthamiana* leaves were first infiltrated with GFP:ATG8CL construct and infected one day later with droplets from a zoospore solution of *P. infestans* as described earlier (**Song et al., 2009**). Water droplets were used as mock treatment. Protein extracts were prepared 2 and 3 days after infection and protein levels of GFP:ATG8CL were detected by immunoblots using a polyclonal GFP-HRP antibody (Santa Cruz, Texas, USA).

## PexRD54/Joka2 *in planta* competition assays for ATG8CL binding

Joka2:RFP was transiently co-expressed with GFP:ATG8CL in the presence of HA:PexRD54, HA:PexRD54[AIM], or HA:EV constructs in *N. benthamiana* leaves. Total proteins extracts prepared 2 days after infiltration were then used in anti-GFP Co-IPs. Purified protein complexes were separated by SDS/PAGE and immunoblotted to detect RFP, GFP and HA signals. Similar competition experiment was conveyed with increased HA:PexRD54 *A. tumefaciens* concentrations ($OD_{600}$=0, 0.05, 0.1, 0.2 or 0.4) while Joka2:RFP and GFP:ATG8CL *A. tumefaciens* concentrations were kept fixed at $OD_{600}$=

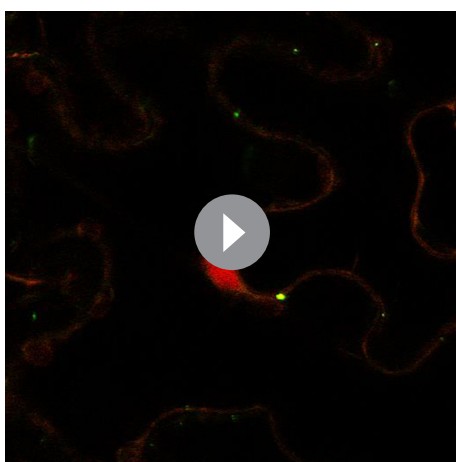

**Video 4.** RFP:ATG8CL and Joka2:GFP colocalize at mobile endomembrane compartments. RFP:ATG8CL was transiently coexpressed with Joka2:GFP in *N. benthamiana* leaves and examined by confocal laser scanning microscopy, 3 days post infiltration. (Related to *Figure 6*)

0.2. Protein extracts prepared two days after infiltration were used in GFP-IPs as described above. Joka2-ATG8CL binding assays were carried out by co-expressing Joka2:RFP and GFP:ATG8CL in the presence of HA:PexRD54, HA:PexRD54$^{AIM2}$, HA:PexRD54$^{J2AIM1}$ (PexRD54 AIM (LDFDWEIV) replaced by Joka2 AIM (GVAEWDPI)), HA:PexRD54$^{J2AIM2}$ (PexRD54 AIM replaced by Joka2 AIM plus 8 additional amino acids at the C-terminus (GVAEWDPILEELKEMG)) or HA:EV. GFP-IPs were performed as described above.

## Infection assays

Infection assays assessing the effect of Joka2 overexpression on *P. infestans* colonization were performed as follow: Joka2:RFP, Joka2$^{AIM}$:RFP or RFP EV were transiently overexpressed side by side on either halves of independent *N. benthamiana* leaves. Twenty-four hours after expression, the infiltrated leaves were detached and inoculated with *P. infestans* 88,069 on two or three spots on each half leaf. *P. infestans* growth was monitored by UV photography 6 days after infection. Colonization was quantified by measuring the diameter of the lesion on each inoculated spot and values from three independent biological replicates were used to generate the scatter plots.

To demonstrate whether PexRD54 could alleviate the effect of Joka2 on pathogen growth, Joka2:RFP was co-expressed with HA:PexRD54, HA:PexRD54$^{AIM2}$ or HA:EV in *N. benthamiana* leaves, which were infected and monitored as described above. For all infection assays, experiments were repeated three times with minimum 10 infection spots. Lesion diameters were measured 6 days after infection.

## Joka2 silencing assays

Virus induced gene silencing of Joka2 was performed in *N. benthamiana* as described previously (*Liu et al., 2002*). Two different TRV2/pYL279 constructs were designed to target both 5'-and 3' ends of Joka2. TRV:Joka2-1 and TRV2:Joka2-2 targeted the region between 340 and 639 and between 1942 and 2241, respectively. The primers were used for generating TRV2:Joka2-1 and TRV2:Joka2-2 are listed in *Supplementary file 3*. TRV2:GFP was used as a negative control as described previously (*Chaparro-Garcia et al., 2015*).

Suspensions of *Agrobacterium tumefaciens* strain *GV3101* harboring TRV1/pYL155 and TRV2:Joka2-1 or TRV2:Joka2-2 were mixed in a 2:1 ratio in infiltration buffer (10 mM MES (2-[N-morpholino]ethanesulfonic acid), 10 mM magnesium chloride (MgCl$_2$), pH 5.6) to a final OD600 of 0.3. As a control, we used TRV2:GFP. Two-week-old *N. benthamiana* plants were infiltrated with *A. tumefaciens* for VIGS assays and upper leaves were used 2–3 weeks later for *P. infestans* infections. UV photographs were taken 5 days post infection. Each experiment was repeated at least three times with minimum 10 infection spots and disease lesion areas were measured using ImageJ. Each lesion size value was normalized following the formula 'Nr = Or * Ar /A', where N is the normalized lesion size value, O is the original lesion size value, r is the biological repeat, Xr is the average of all O values of the biological repeat r, and A is the average of all O values. Silencing levels were confirmed using RT-PCR. Total RNA was extracted using RNAeasy Plant Mini Kit (Qiagen, UK) and treated with Ambion TURBO DNA-*free* according to manufacturer's protocol. 1.0 μg of DNase treated RNA was used for cDNA synthesis using SuperScript III Reverse Transcriptase (Invitrogen). RT-PCR was performed with the following program: 1 cycle with 3 min at 95℃, followed by 35 cycles with 95℃ at 30 s, 56℃ at 30 s and 72℃ at 30 s. Primers pairs used for cDNA amplification were Joka2-TRV1-F

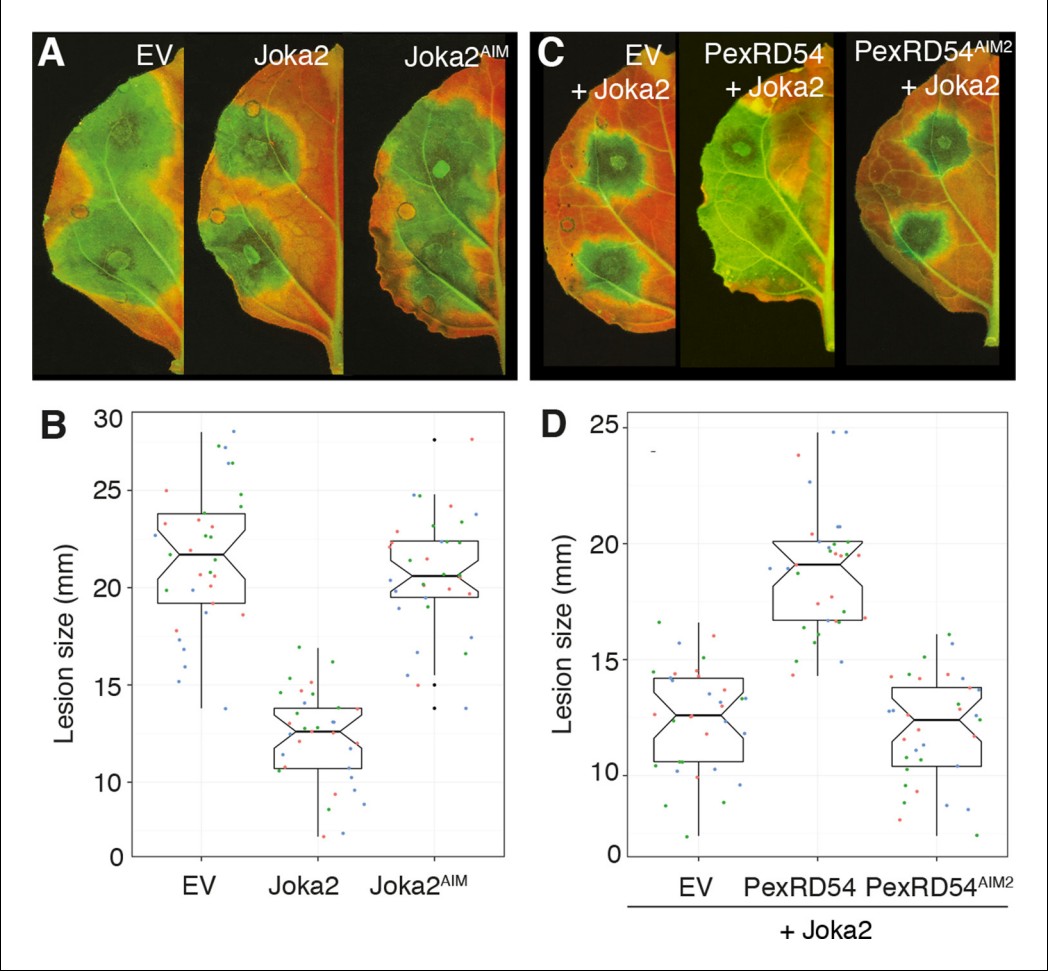

**Figure 7.** PexRD54 counteracts the enhanced resistance conferred by Joka2. (**A**) Overexpression of Joka2:RFP limits *P. infestans* colonization. Halves of *N. benthamiana* leaves expressing RFP:EV, Joka2:RFP and Joka2AIM:RFP were infected with *P. infestans* and pathogen growth was determined by lesion sizes measured 6 days post-inoculation. (**B**) Categorical scatter plots show lesion diameters of 11 infections sites from three independent biological replicates pointed out by three different colors. Similar p values (p<0.001) were obtained in three independent biological repeats. (**C**) PexRD54 counteracts the effect of Joka2 on *P. infestans* colonization. Joka2: RFP was co-expressed with HA:EV, HA:PexRD54 or HA:PexRD54AIM2 in *N. benthamiana* leaves which are then inoculated with *P. infestans*. Joka2:RFP failed to limit pathogen growth in the presence of PexRD54, whereas it could still restrict pathogen growth in the presence of PexRD54AIM2 or vector control (EV). (**D**) Categorical scatter plots show lesion diameters of 11 infections sites from three independent biological replicates pointed out by three different colors. Similar p values (p<0.001) were obtained in three independent biological repeats.

The following figure supplement is available for figure 7:

**Figure supplement 1.** Silencing of Joka2 enhances susceptibility to *P. infestans*.

and Joka2-TRV1-R, and Joka2-TRV2-R and Joka2-TRV2-F (described in *Supplementary file 3*). *NbEF1α* was used to normalize transcript abundance (*Segonzac et al., 2011*).

## Live-cell imaging by confocal laser scanning microscopy

All microscopy analyses were performed on live leaf tissue 3 days post agroinfiltration. *N. benthamiana* leaf patches were cut and mounted in water and analyzed on a Leica TCS SP5 confocal microscope (Leica Microsystems, Germany) using 63x water immersion objective. The GFP and RFP probes were excited using 488 and 561 nm laser diodes and their fluorescent emissions were collected at 495–550 nm and 570–620 nm, respectively. To avoid bleed-through from different

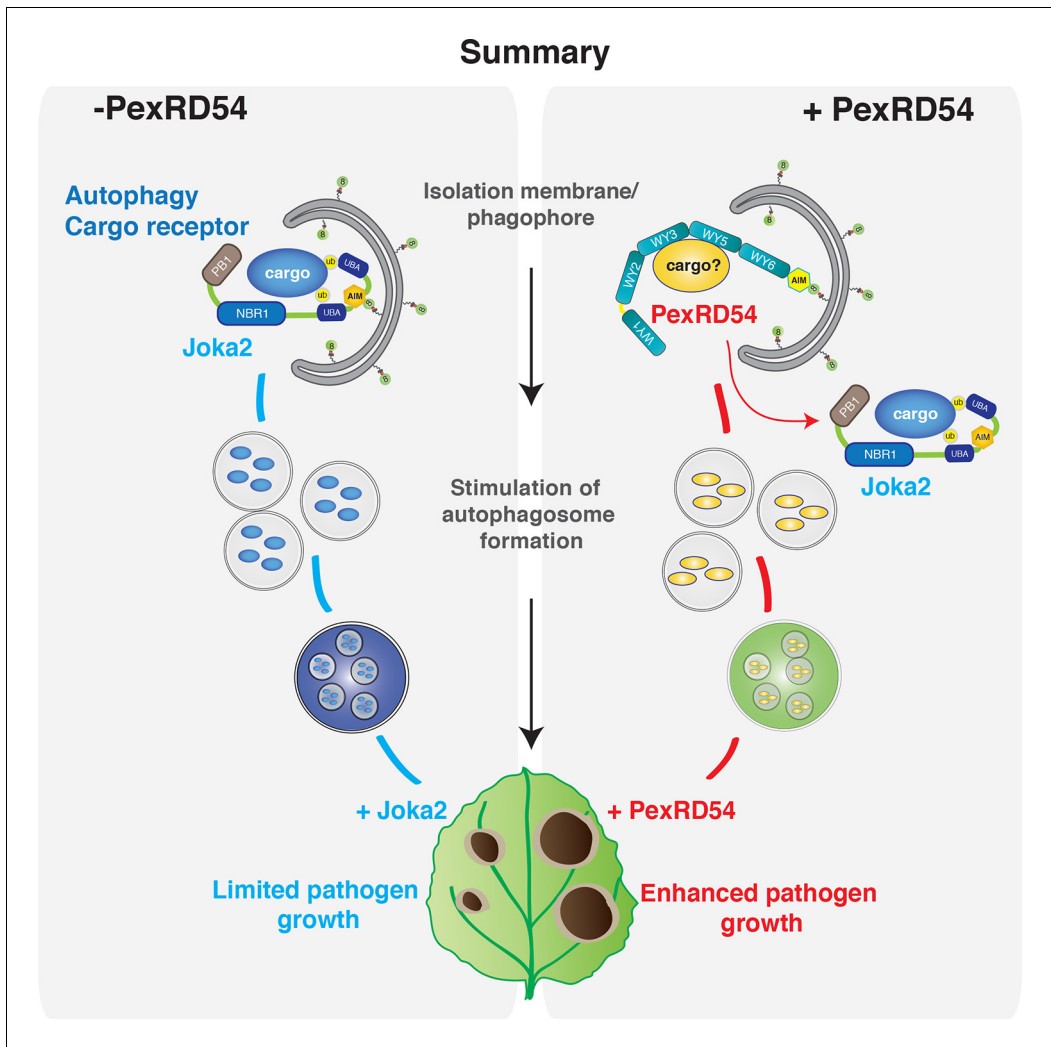

**Figure 8.** PexRD54 outcompetes the autophagy cargo receptor Joka2 and enhances virulence.

fluorophores, co-localization images were taken using sequential scanning between lines and acquired using multi-channels.

## Image processing and data analysis

Confocal microscopy images were processed with the Leica LAS AF software, ImageJ (2.0) and Adobe PHOTOSHOP CS5 (12.0) programs. Images for quantification of autophagosome numbers were obtained from 50 Z stacks consisting of 1 μm depth field multi-layered images with similar settings for all samples. To detect and quantify punctate structures in one channel (green channel or red channel or overlay channel (green channel images superimposed with red channel ones)), the Z stacks were separated into individual images with the ImageJ (2.0) program and analyzed. The counting procedure was based on a naked-eye detection of punctate structures to avoid cytoplasm noise and dual counting of autophagosomes within the same stack. Histograms were generated with mean of punctate numbers generated from stacks obtained in two independent biological experiments. Statistical differences were assessed by means of a two-tailed t-test assuming unequal variance as implemented in StatPlus LE package (AnalystSoft, Washington, USA). Measurements were significant when $p < 0.05$ and highly significant when $p < 0.001$.

## Electron microscopy and immunogold labelling

Leaf samples were embedded in LR White as described (*Liu et al., 2002*) except sections were picked up on gold grids before immunogold labeling. All labeling procedures were carried out at room temperature. Grids were floated, section-side down, on drops of 50 mM glycine/PBS (150 mM NaCl, 10 mM phosphate, pH 7.4) for 15 min then on Aurion blocking buffer (5% BSA/0.1% cold water fish gelatin/5–10% normal goat serum/15 mM NaN₃/PBS, pH7.4) (Aurion, the Netherlands) for 30 min then briefly equilibrated in incubation buffer (0.1% (v/v) BSA-C (actetylated BSA; Aurion) / PBS, pH 7.3) before a 90 min' incubation with the primary antibody. Serial sections on separate grids were labelled with either a rabbit polyclonal anti-GFP Abcam ab6556 (Abcam, UK) at 1/250, or a rabbit polyclonal anti-catalase AS09 501 (Newmarket Scientific, UK) at 1/1000. Grids were washed 6x5 min in drops of incubation buffer then placed on the drops of goat anti-rabbit secondary antibody conjugated to 10 nm gold (BioCell, Agar Scientific Ltd., Essex, UK), diluted to 1/50 in incubation buffer, for 90 min. After 6x5 min washes in incubation buffer and 2x5 min washes in PBS, the grids were briefly washed in water and contrast stained with 2% (w/v) uranyl acetate. Grids were viewed in a FEI Tecnai 20 transmission electron microscope (FEI, the Netherlands) at 200 kV and digital TIFF images were taken using an AMT XR60B digital camera (Deben, UK) to record TIFF files. Bar charts were generated by counting the clustered (>4 gold particles close to each other) gold particles on each image.

## Chemical treatments

3-Methyl adenine (3-MA), a phosphatidylinositol 3-kinase (PI3K) inhibitor is widely used to inhibit autophagosome formation (*Hanamata et al., 2013*). *N. benthamiana* leaves transiently expressing GFP:ATG8CL-RFP:PexRD54, GFP:PexRD54-RFP:ATG8CL or YFP:VTI12 were infiltrated with 5 mM 3-MA. Both GFP:PexRD54 and RFP:ATG8CL constructs were expressed side by side to monitor the effect of 3-MA treatment on stimulation of autophagosome formation by PexRD54. Punctate structures were visualized using confocal microscopy 6–10 hr after 3-MA treatment. Bar charts were generated with the number of punctate structures obtained from maximum projections of Z-stack images of two independent biological experiments.

The cysteine protease inhibitor E64d is widely used for measuring autophagic flux (*Bassham, 2015*). To determine whether PexRD54 blocked autophagic flux, at 2 dpi we infiltrated leaves with 100 μM E64D and kept them overnight in the dark. At 3 dpi, we collected E64d treated and untreated samples and analyzed total protein levels using appropriate antisera. Concanamycin A (2 μM in agro infiltration medium) was infiltrated into leaves of *N. benthamiana* transiently expressing ATG8CL and PexRD54, PexRD54[AIM2] or empty vector control constructs. The leaves were than incubated in dark at 20°C for 24 hr. ATG8CL-labelled puncta were visualized using confocal microscopy 24 hr after concanamycin A treatment.

## Cloning, expression and purification of PexRD54 and PexRD54[AIM2] variants for in vitro studies

DNA encoding PexRD54 residues Val92 to Val381 (lacking secretion and translocation signals) was amplified from RFP:PexRD54 (using primers shown in *Supplementary file 3*) and cloned into the vector pOPINS3C, resulting in an N-terminal 6xHis-SUMO tag with PexRD54, linked by a 3C cleavage site (*Berrow et al., 2007*). Recombinant protein was produced using *E. coli* BL21-Arabinose Inducible (AI) cells. Cell cultures were grown in Power Broth at 37°C to an $A_{650}$ 0.4–0.6 followed by induction with 0.2% (w/v) L-arabinose and overnight incubation at 18°C. Pelleted cells were resuspended in buffer A (50 mM Tris-HCl pH 8, 500 mM NaCl, 50 mM glycine, 5% (v/v) glycerol and 20 mM imidazole supplemented with EDTA free protease inhibitor tablets (one tablet per 40 ml buffer)) and lysed by sonication. The clarified cell lysate was applied to a Ni²⁺-NTA column connected to an AKTA Xpress system. 6xHis-SUMO-PexRD54 was step-eluted with elution buffer (buffer A containing 500 mM imidazole) and directly injected onto a Superdex 75 26/600 gel filtration column pre-equilibrated in buffer C (20 mM HEPES pH 7.5, 150 mM NaCl). The fractions containing 6xHis-SUMO-PexRD54 were pooled and concentrated to 2–3 mg/mL. The 6xHis-SUMO tag was cleaved by addition of 3C protease (10 μg/mg fusion protein) and incubated at 4°C overnight. Cleaved PexRD54 was further purified using Ni²⁺-NTA column (collecting eluate) followed by gel filtration. The purified

protein was concentrated as appropriate, and the final concentration was judged by absorbance at 280 nm (using a calculated molar extinction coefficient of PexRD54, 57,040 $M^{-1}cm^{-1}$).

The PexRD54$^{AIM2}$ variant was amplified from RFP:RD54$^{AIM2}$ (using primers shown in *Supplementary file 3*) and cloned into the vector pOPINS3C as above. Recombinant protein was expressed and purified as for wild-type PexRD54; with protein concentration measured using a calculated molar extinction coefficient of 51,350 $M^{-1}cm^{-1}$.

## Cloning, expression and purification of ATG8CL and ATG8IL

DNA encoding Met1 to Ser119 of ATG8CL and Gly2 to Ser119 of ATG8IL were amplified from GFP:ATG8CL and GFP:ATG8IL (using primers shown in *Supplementary file 3*) and cloned into the vector pOPINF, generating a cleavable N-terminal 6xHis-tag with ATG8CL and ATG8IL. Recombinant proteins were produced using *E. coli* strain BL21 (DE3) grown in lysogeny broth at 37°C to an $A_{600}$ of 0.4–0.6 followed by induction with 1 mM IPTG and overnight incubation at 18°C. Pelleted cells were resuspended in buffer A and pure, concentrated protein prepared as described for PexRD54 above (concentration determined using a calculated molar extinction coefficient of 7680 $M^{-1}cm^{-1}$ and 9080 $M^{-1}cm^{-1}$ for ATG8CL and ATG8IL, respectively).

## Isothermal titration calorimetry (ITC)

Calorimetry experiments were carried out at 15°C in 20 mM HEPES pH 7.5, 500 mM NaCl, using an iTC200 instrument (MicroCal Inc.). For protein:protein interactions, the calorimetric cell was filled with 100 µM PexRD54 and titrated with 1.1 mM ATG8CL or ATG8IL from the syringe. A single injection of 0.5 µl of ATG8CL or ATG8IL was followed by 19 injections of 2 µl each. Injections were made at 150 s intervals with a stirring speed of 750 rpm. For the heats of dilution control experiments, equivalent volumes of ATG8CL or ATG8IL were injected into buffer using the parameters above. For protein:peptide interactions, the calorimetric cell was filled with 90 µM ATG8CL or ATG8IL and titrated with 1 mM peptide from the syringe. The titrations were performed at 25°C, but otherwise as above. The raw titration data were integrated and fitted to a one-site binding model using the MicroCal Origin software.

## Accession numbers

PexRD54 (PITG_09316); *St*ATG8CL (PGSC0003DMP400038670), *St*ATG8IL (PGSC0003DMP400009229), *Sl*ATG8CL (Solyc10g006270 and Solyc07g064680), *Sl*ATG8IL (Solyc01g068060), *Nb*ATG8CL (Nb_S00003316g0005, KR021366), *Nb*ATG8IL (Nb_S00005942g0011, KR021365), and Joka2 (XM_006344410).

## Acknowledgements

We thank Elaine Barclay, Liliana M. Cano, Frédéric Fragnière, Marina Franceschetti, Artemis Giannakopoulou, Ricardo Oliva, Egem Ozbudak, Stephen Whisson for technical support and/or providing materials. We are grateful to Zlay Taftacs and all members of the Kamoun Lab for helpful suggestions. This project was funded by the Gatsby Charitable Foundation, European Research Council (ERC), Biotechnology Biological Sciences Research Council (BBSRC) and the John Innes Foundation.

## Additional information

### Funding

| Funder | Author |
| --- | --- |
| European Research Council | Abbas Maqbool<br>Benjamin Petre<br>Joe Win<br>Sophien Kamoun |
| Gatsby Charitable Foundation | Yasin F Dagdas<br>Khaoula Belhaj<br>Neftaly Cruz-Mireles<br>Jan Sklenar<br>Joe Win |

|  | Frank Menke |
|  | Sophien Kamoun |
| Biotechnology and Biological Sciences Research Council | Pooja Pandey |
|  | Nadra Tabassum |
|  | Tolga O Bozkurt |

The funders had no role in study design, data collection and interpretation, or the decision to submit the work for publication.

### Author contributions

YFD, KB, AM, TOB, Conception and design, Acquisition of data, Analysis and interpretation of data, Drafting or revising the article; AC-G, BP, JS, JW, KF, Acquisition of data, Analysis and interpretation of data, Drafting or revising the article; PP, NT, NC-M, RKH, Conception and design, Acquisition of data, Analysis and interpretation of data; FM, MJB, SK, Conception and design, Analysis and interpretation of data, Drafting or revising the article

### Author ORCIDs

Yasin F Dagdas, http://orcid.org/0000-0002-9502-355X
Nadra Tabassum, http://orcid.org/0000-0002-2722-9441
Mark J Banfield, http://orcid.org/0000-0001-8921-3835
Sophien Kamoun, http://orcid.org/0000-0002-0290-0315

## Additional files

### Supplementary files

• Supplementary file 1. Plant proteins that associate with PexRD54 as identified by mass spectrometry after immunoprecipitation (IP). FLAG:PexRD54 was transiently expressed in *N. benthamiana* leaves and proteins were extracted two days after infiltration and used in IP experiments. Unique spectral counts are given for each control and PexRD54 sample. Proteins related to vesicle trafficking and autophagy are highlighted in yellow. ATG8CL is highlighted in blue and found in several replicates including the stringent IP experiment.

• Supplementary file 2. Unique peptides obtained from liquid chromatography tandem mass spectrometry (LC-MS/MS) analysis of PexRD54 immunoprecipitation experiments suggest specific association with ATG8CL. Although ATG8 protein family is highly conserved, and there are eight ATG8 variants in *N. benthamiana*, unique peptides obtained from LC-MS/MS analysis enabled specific identification of ATG8CL variant (Nb_S00003316g0005) from *N. benthamiana*. ATG8CL variant of *N. benthamiana* has identical amino acid sequence with the potato ATG8CL variant that is used for other experiments.

• Supplementary file 3. Primers used in this study.

### Major datasets

The following datasets were generated:

| Author(s) | Year | Dataset title | Dataset URL | Database, license, and accessibility information |
|---|---|---|---|---|
| Nusbaum C, Haas B, Kamoun S, Fry W, Judelson H, Ristaino J, Govers F, Whis- son S, Birch P, Birren B, Lander E, Gala- gan J, Zody M, De- von K, O'Neil K, Zem- bek L, Ander- son S, Jaffe D, But- ler J, Alvarez P, Gnerre S, Grabherr M, Mauceli E, Brockman W, Young S, LaButti K, Sykes S, DeCaprio D, Craw- ford M, Koehrsen M, Engels R, Mon- tgomery P, Pearson M, Ho- warth C, Lar- son L, White J, O'Leary S, Kodira C, Zeng Q, Yandava C, Alvara- do L | 2006 | Phytophthora infestans T30-4 secreted RxLR effector peptide protein, putative (PITG_09316) mRNA, complete cds | http://www.ncbi.nlm.nih. gov/nuccore/XM_ 002903553.1 | Publicly available at NCBI Nucleotide (Accession no: XM_ 002903553.1) |
| Fernandez-Pozo et al | 2014 | Solanum tuberosum polypeptide PGSC0003DMP400038670 | https://solgenomics.net/ feature/19557200/details | Publicly available at SolGenomics Database (Accession no: PGSC0003 DMP400038670) |
| Fernandez-Pozo et al | 2014 | Solanum tuberosum polypeptide PGSC0003DMP400009229 | https://solgenomics.net/ feature/19596338/details | Publicly available at SolGenomics Database (Accession no: PGSC0003 DMP400009229) |
| Fernandez-Pozo et al | 2010 | Tomato locus Solyc10g006270 | https://solgenomics.net/ locus/35795/view | Publicly available at SolGenomics Database (Accession no: Solyc10g006270) |
| Fernandez-Pozo et al | 2010 | Tomato locus Solyc07g064680 | https://solgenomics.net/ locus/30483/view | Publicly available at SolGenomics Database (Accession no: Solyc07g064680) |
| Fernandez-Pozo et al | 2010 | Tomato locus Solyc01g068060 | https://solgenomics.net/ locus/11195/view | Publicly available at SolGenomics Database (Accession no: Solyc01g068060) |
| Dagdas YF, Belhaj K, Maqbool A, Chaparro-Garcia A, Pandey P, Petre B, Tabassum N, Cruz-Mireles N, Hughes RK, Sklenar J, Win J, Menke F, Findlay K, Banfield MJ, Kamoun S, Bozkurt TO | 2016 | An effector of the Irish potato famine pathogen antagonizes a host autophagy cargo receptor | https://www.ncbi.nlm. nih.gov/nuccore/ 976151098/ | Publicly available at NCBI (Accession no: KR021366) |

| Dagdas YF, Belhaj K, Maqbool A, Chaparro-Garcia A, Pandey P, Petre B, Tabassum N, Cruz-Mireles N, Hughes RK, Sklenar J, Win J, Menke F, Findlay K, Banfield MJ, Kamoun S, Bozkurt TO | 2015 | An effector of the Irish potato famine pathogen antagonizes a host autophagy cargo receptor | https://www.ncbi.nlm.nih.gov/nuccore/976151096/ | Publicly available at NCBI (Accession no: KR021365) |
| Dagdas YF, Belhaj K, Maqbool A, Chaparro-Garcia A, Pandey P, Petre B, Tabassum N, Cruz-Mireles N, Hughes RK, Sklenar J, Win J, Menke F, Findlay K, Banfield MJ, Kamoun S, Bozkurt TO | 2016 | PREDICTED: Solanum tuberosum protein NBR1 homolog (LOC102582603), transcript variant X2, mRNA | https://www.ncbi.nlm.nih.gov/nuccore/971545348/ | Publicly available at NCBI nucleotide (Accession no: XM_006344410.2) |
| Dagdas YF, Belhaj K, Maqbool A, Cha- parro-Garcia A, Pandey P, Petre B, Tabassum N, Cruz- Mireles N, Hughes RK, Sklenar J, Win J, Menke F, Findlay K, Banfield MJ, Ka- moun S, Bozkurt TO | 2016 | Data from: An effector of the Irish potato famine pathogen antagonizes a host autophagy cargo receptor | http://dx.doi.org/10.5061/dryad.dv56j | Available at Dryad Digital Repository under a CC0 Public Domain Dedication. |

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
