## [Decision Letter]

Thank you for submitting your work entitled "An effector of the Irish potato famine pathogen antagonizes a host autophagy cargo receptor" for consideration by *eLife*. Your article has been reviewed by three peer reviewers, and the evaluation has been overseen by Jean Greenberg (Reviewing Editor) and Detlef Weigel (Senior Editor).

The reviewers have discussed the reviews with one another and the Reviewing Editor has drafted this decision to help you focus on the additional work we consider essential for a favorable decision on this submission.

Summary:

This manuscript presents exciting data suggesting that the *P. infestans* effector PexRD54 targets the autophagy system and may affect autophagy through a competition mechanism. We appreciate the rigor of the work presented and its clarity of exposition. However, we think there are important issues that require additional work, which is explained below.

Essential revisions:

1) The quality of the TEM experiment:

The TEM data in the manuscript show somewhat different localization compared to the confocal images. For example, the confocal image shows GFP-tagged PexRD54 mainly localized in some foci and the cytosol (Figure 3—figure supplement 1), but the TEM image (Figure 3—figure supplement 2, please correct the figure legend, which explains A and B but not panel C) does not show any particles in the cytosol. Moreover the vesicles in which particles accumulate do not have the appearance of autophagosomes (they seem more like an aggresome due to overexpression) and this cannot explain the ring-like structure of PexRD54 (Figure 4). There should be a good control photo showing an autophagosome, probably marked with ATG8CL. We suggest observing the sample expressing both constructs of PexRD54 and ATG8CL for TEM to see both at the autophagosome. Since the autophagosomes induced by PexRD54 have altered morphology (compare Figure 4 to Figure 6—figure supplement 3), please address the morphology difference in the manuscript. Differences in morphology could also affect quantification. Are the differences the same when overlap is quantified in a 50x50 micron area compared to all puncta found in the area? The TEM data is very important and should be in the main figure. Instead, Figure 2 can be in the supplement.

2) Biological relevance:

A weakness of the work is a lack of information about the relevance of autophagy for this pathosystem and the relevance of PexRD54 for virulence. We think these issues can be addressed by: (1) silencing PexRD54 to address its role in virulence and its effect on autophagy during infection; and/or (2) performing an experiment to test the importance of autophagy for this pathosystem.

Additional critique:

You should address the following comments:

3) Figure 3—figure supplement 4 shows expression of GFP-ATG8CL, with only about 50 puncta/10,000 µm^2, while Figure 4 counts about 10 puncta/10,000 µm^2 (with EV control). Please explain the discrepancy. It is important to address this as you claim that PexRD54 enhances autophagy activity based on the ATG8CL counts (about 40 puncta/10,000 µm^2 in Figure 4).

4) Can you assay for autophagy flux biochemically to bolster your conclusion thatPexRD54 is affecting autophagy flux?

5) Figure 2 – it seems that AVRBlb2 interacts weakly with ATG8IL. Does it have an AIM domain or is ATG8IL a bit sticky? Can you please quantify binding in the protein blot?

6) All experiments are performed using transient overexpression in *Nicotiana*. Please comment on the caveats with this approach and consider doing some validation using native promoters to validate findings in a more biologically relevant context.

7) Since PexRD54 can stimulate the formation of ATG8CL-marked autophagosomes, you suggest that PexRD54 can activate autophagy for the pathogen's own benefit (Pro-death?). You conclude at the same time that PexRD54 counteracts the positive role of Joka2-mediated selective autophagy in pathogen defense (Pro-survival?). Is it either or both? Please clarify.

[Editors' note: further revisions were requested prior to acceptance, as described below.]

Thank you for resubmitting your work entitled "An effector of the Irish potato famine pathogen antagonizes a host autophagy cargo receptor" for further consideration at *eLife*. Your revised article has been favorably evaluated by Detlef Weigel (Senior editor), a Reviewing editor, and four reviewers. The manuscript has been improved, but there are some minor remaining issues that need to be addressed before acceptance, as outlined below:

The work is substantially improved. An EM expert has suggested a number of improvements that we would like you to implement (none of which require additional experimentation):

1) About Figure 3—figure supplement 3:

The upper and lower photos seem to have different magnification. Please add size bars in the upper images.

Legends: "Vesicles labelled with GFP antibody were different than the vesicles labelled with catalase antibody, confirming that PexRD54 labelled vesicles are not peroxisomes." Please do not use the term "vesicles" here, since it is not sure that they are not vesicles. Instead, the description can be considered as: high electron dense regions (structures)

Also the statement: "Stars indicate vesicular structures that are labelled by gold particles in the other image" is improper. The stars in two images obviously indicate different structures- the upper image indicates electron-dense structure and the lower image indicates peroxisomes (organelle). It is better to use different labels.

For anti-GFP, the upper image indicates that gold particles were labeled in the electron-dense portion, which was neither associated with vesicle-like structures nor peroxisomes.

2) About Figure 3—figure supplement 2:

Based on the total set of supplemental EM images which authors provided, Figure 3—figure supplement 2 seems not to be a typical image. Since most of images do not have vesicle like structures, the photo of number 11 seems to represent the total set of images.

3) Methods – Electron Microscopy and Immunogold labelling:

Since authors used high pressure freezing to get immunogold labelling, the authors need to describe the process in the Methods. I think authors may have cited the wrong reference (Liu, Schiff and Dinesh-Kumar et al., 2002), please check it.

In your revision, please include your archive of EM images as supplemental material so that readers can see the whole range of data.

Major reviewer comments were:

Reviewer #1:

I think the authors improved the manuscript. The effector biology has a limited way of doing experiments but this manuscript dealt with it with a lot of controls, thus convincing.

Reviewer #2:

The revised manuscript of Dagdas and colleagues report the novel finding that an oomycete effector (PexRD54) binds to a plant host autophagy protein (ATG8CL) to trigger autophagosome production. As a result plant defense are circumvented. The original manuscript was extensively reviewed and in general, this exciting paper was well received. However, there were two major points of concern:

1) TEM-image quality

2) Biological Relevance

My main concern was point #2; biological relevance. In my view, while this is an excellent study with numerous nuggets that will be of interest to the *eLife* community. This work would be significantly compromised without some discussion, experimentation to indicate that autophagy is relevant to this pathosystem.

Briefly, to demonstrate the importance of autophagy in this system the cargo receptor Joka2 was silenced and oomycete colonization, measured using two distinct constructs; thus strengthening the argument outlined in Figure 7 and Figure 7—figure supplement 1; conclusions indicating the importance of selective autophagy in this pathosystem.

The authors also discuss the very real technical difficulties in this system and the relatively high levels of variation that complicate interpretation of these studies/this system has a high degree of difficulty in and of itself.

In summary, by performing silencing experiments and demonstrating, the importance of autophagy regulation in this system, it is my view that the authors have adequately met the reviewers’ concerns regarding biological relevance.

Reviewer #3:

In this revision, the authors have addressed some of the previous concerns about the manuscript. My primary concern previously was about biological relevance due to experiments using overexpression in *Nicotiana*. To address biological relevance, Joka2 was silenced using VIGS in *Nicotiana* and demonstrated enhanced susceptibility to *P. infestans*. These results are consistent with the observation that overexpression of Joka2 enhances resistance. The authors provide a detailed rebuttal to why they have not performed silencing in *P. infestans*. While I would like to see the field move away from primarily relying on overexpression to deduce phenotypes, this manuscript does present significant insight into the role of effector-modification of autophagy and is worthy of publication.

Reviewer #4:

In this revised version, in response to the reviewers’ suggestions, the authors have done TEM on serial sections and confirmed that anti-catalase conjugated gold particles did not co-localize with GFP coated gold particles, indicating that cells expressing GFP:PexRD54 revealed a strong signal in certain portions that are not peroxisomes.

The movies are a good evidence to confirm that the mobile structures are vesicles.

---

## [Author Response]

Essential revisions:

1) The quality of the TEM experiment:

The TEM data in the manuscript show somewhat different localization compared to the confocal images. For example, the confocal image shows GFP-tagged PexRD54 mainly localized in some foci and the cytosol (Figure 3—figure supplement 1), but the TEM image (Figure 3—figure supplement 2, please correct the figure legend, which explains A and B but not panel C) does not show any particles in the cytosol. Moreover the vesicles in which particles accumulate do not have the appearance of autophagosomes (they seem more like an aggresome due to overexpression) and this cannot explain the ring-like structure of PexRD54 (Figure 4). There should be a good control photo showing an autophagosome, probably marked with ATG8CL. We suggest observing the sample expressing both constructs of PexRD54 and ATG8CL for TEM to see both at the autophagosome. Since the autophagosomes induced by PexRD54 have altered morphology (compare Figure 4 to Figure 6—figure supplement 3), please address the morphology difference in the manuscript. Differences in morphology could also affect quantification. Are the differences the same when overlap is quantified in a 50x50 micron area compared to all puncta found in the area? The TEM data is very important and should be in the main figure. Instead, Figure 2 can be in the supplement.

We agree that the provided image is not conclusive on its own regarding the autophagosome localization of the effector. Because of that we were very careful not to use the term autophagosome to define the observed localization pattern. This is what we said in the manuscript: “Immunogold labeling in transmission electron micrographs of cells expressing GFP:PexRD54 revealed a strong signal in vesicular compartments”. Our most conclusive experiments with regards to autophagosome localization of PexRD54 are (i) autophagy inhibitor 3-methyl adenine, (ii) confocal live cell imaging, (iii) ATG8 terminal glycine dominant negative mutant. We are also confident that PexRD54 labeled endomembrane compartments are not aggresomes, because unlike the protein aggregates they are mobile (Video 1). We have changed the electron micrograph image that we used in the figure. We believe the current image delivers our message clearly. Additionally we are providing a selection of images as a supplementary data set. These images were used to create the bar chart in Figure 3—figure supplement 2 B and they clearly demonstrate accumulation of PexRD54 at vesicles.

As the reviewers suggested we have done TEM on serial sections of GFP-PexRD54 expressing cells to see if the vesicles we observed are peroxisomes. As you can clearly see in Figure 3—figure supplement 3, anti- catalase conjugated gold particles did not co-localize with GFP coated gold particles. We can now conclude that GFP-PexRD54 accumulates in vesicles that are not peroxisomes.

We have tried to do serial sectioning TEM using gold conjugated ATG8 antibodies. We used two different ATG8 antibodies: (i) AS14 2769 from Agrisera raised for *Chlamydomanas* ATG8; (ii) ab77003 from abcam raised for yeast ATG8 protein. These antibodies were suggested to us by members of the plant autophagy community. Unfortunately none of these antibodies were specific in TEM trials and we could not proceed further with these experiments. Nonetheless as we stated above, we believe we have several lines of evidence confirming that PexRD54 localize at ATG8CL labeled autophagosomes.

It was noted that the PexRD54 labeled vesicles are not surrounded by double membranes. Please note that membranes cannot be visualized in the TEM experiments we conducted because we have used high pressure freezing to get better immunogold labelling.

2) Biological relevance:

A weakness of the work is a lack of information about the relevance of autophagy for this pathosystem and the relevance of PexRD54 for virulence. We think these issues can be addressed by: (1) silencing PexRD54 to address its role in virulence and its effect on autophagy during infection; and/or (2) performing an experiment to test the importance of autophagy for this pathosystem.

To further demonstrate that autophagy is important for this pathosystem, we complemented the original experiments with silencing the autophagy cargo receptor, Joka2 and measured *P. infestans* colonization in silenced plants. Using two different silencing constructs we have shown that silencing of Joka2 enhances susceptibility to *P. infestans*. These results are now presented in Figure 7—figure supplement 1. Our silencing results are consistent with Joka2 overexpression phenotype presented in Figure 7, where we see enhanced resistance in Joka2 overexpressing leaves. These experiments clearly demonstrate the relevance of selective autophagy in the *P. infestans* pathosystem. Furthermore this is the first report demonstrating that selective autophagy and the cargo receptor Joka2 play a positive role in antimicrobial immunity in plants.

The reliance on in planta expression is a recurring issue in effector biology. Nonetheless, much progress has still been made in this field even with obligate pathogens where there is no chance of having knock-outs or knock-downs. Unfortunately, our *P. infestans* system is simply not reliable enough for gene silencing as detailed below. This is counteracted by using mutants of the effector that can genetically link various phenotypes, such as binding to a host interactor, effect on virulence, etc.

There are technical difficulties of gene silencing in *P. infestans*, which is also much harder to transform than a couple other *Phytophthora* spp. An important issue with *P. infestans* transformation is phenotypic variation between transformants, which confounds interpretation of the results. Transformants of the same construct are variable in virulence, growth and habit, and also are typically unstable. Even empty vector control transformants can show defects in virulence. Considering the expected quantitative phenotypes, this variation may mask the potential phenotype of PexRD54 silencing. Also, only a few labs managed to stably silence genes in *P. infestans* and results are typically not readily reproducible, even within individual labs. Recently, a new issue popped up because the changes in heterochromatin that are associated with gene silencing appear to affect other (linked or unlinked) genes. Yet, another argument against the robustness of the method. Although we are trying to improve our methods to manipulate *P. infestans* (CRISPR etc), so far we did not manage to get a reproducible knock-out/knock-down system.

To overcome this, we used the effector mutant “AIM2” as a control in every experiment. Although AIM2 mutant is only 2 amino acids different from PexRD54 and is equally stable, it lost its ability to bind its host target ATG8, failed to stimulate autophagy, and failed to subvert autophagy related defenses. We believe the AIM2 mutant ensures that the observed phenotypes are not artifacts of the transient expression system. This mutant was used as a negative control in ALL functional assays and provided a genetic link between multiple experimental readouts. We feel using effector mutants is a critical check in effector biology to ensure the validity of the interactors and other readouts.

Note also that in most systems (even with *Pseudomonas syringae* and animal pathogens, see for example Galán J. Cell Host Microbe.5, 571, 2009), knock outs of effector genes typically do not reveal phenotype probably due to redundancy. Thus PexRD54 silencing is unlikely to yield mechanistic insights on manipulation of autophagy by *P. infestans*.

Additional critique:

You should address the following comments:

3) Figure 3—figure supplement 4 shows expression of GFP-ATG8CL, with only about 50 puncta/10,000 µm^2, while Figure 4 counts about 10 puncta/10,000 µm^2 (with EV control). Please explain the discrepancy. It is important to address this as you claim that PexRD54 enhances autophagy activity based on the ATG8CL counts (about 40 puncta/10,000 µm^2 in Figure 4).

We thank the reviewers for raising this point and we are sorry for the confusion. As we wrote in the Materials and methods part (Chemical treatments), in the 3-MA treatment we boosted the number of autophagosomes by coexpressing GFP:PexRD54 and GFP:ATG8CL with RFP:ATG8CL and RFP:PexRD54, respectively, to make sure that the phenotype we are observing with the inhibitor treatment is more clear. We have now modified the figure and figure legend to make this clear.

4) Can you assay for autophagy flux biochemically to bolster your conclusion thatPexRD54 is affecting autophagy flux?

Actually we did not conclude that PexRD54 affects autophagic flux:

“Next we set out to determine the effect of PexRD54 on autophagic flux. […] PexRD54 did not alter the accumulation of GFP:ATG8IL or control GFP protein, confirming that PexRD54 increases ATG8CL protein accumulation specifically (Figure 5).“

5) Figure 2 – it seems that AVRBlb2 interacts weakly with ATG8IL. Does it have an AIM domain or is ATG8IL a bit sticky? Can you please quantify binding in the protein blot?

We thank the reviewers for carefully investigating our results. The band that we see in GFP-IP RFP WB is an unspecific band, because the size of AVRBlb2 is smaller than PexRD54. Also for RFP-IP, we only see a band in saturated exposure conditions and we believe it is not specific. Also, this weak band is not visible in replicate experiments. Additionally there is no predicted AIM motif in AVRBlb2.

6) All experiments are performed using transient overexpression in Nicotiana. Please comment on the caveats with this approach and consider doing some validation using native promoters to validate findings in a more biologically relevant context.

This is a standard approach in effector biology. As we discussed above the use of mutants as negative controls is an important check for linking the different readouts. All experiments used effector mutants as additional controls (in addition to empty vector controls) and we carefully quantified the relative differences based on multiple experimental readouts.

7) Since PexRD54 can stimulate the formation of ATG8CL-marked autophagosomes, you suggest that PexRD54 can activate autophagy for the pathogen's own benefit (Pro-death?). You conclude at the same time that PexRD54 counteracts the positive role of Joka2-mediated selective autophagy in pathogen defense (Pro-survival?). Is it either or both? Please clarify.

We thank the reviewers for the detailed analysis of our results. Please note that we do not know whether there is a link between ATG8CL/JOKA2-mediated selective autophagy and cell death. We believe these terms are not necessarily applicable to our findings but indeed by reviewers’ definition PexRD54 has both pro-death and pro-survival roles.

[Editors' note: further revisions were requested prior to acceptance, as described below.]

The work is substantially improved. An EM expert has suggested a number of improvements that we would like you to implement (none of which require additional experimentation):

1) About Figure 3—figure supplement 3:

The upper and lower photos seem to have different magnification. Please add size bars in the upper images.

We thank the reviewers for pointing this out. We have added the scale bars.

Legends: "Vesicles labelled with GFP antibody were different than the vesicles labelled with catalase antibody, confirming that PexRD54 labelled vesicles are not peroxisomes." Please do not use the term "vesicles" here, since it is not sure that they are not vesicles. Instead, the description can be considered as: high electron dense regions (structures)

Also the statement: "Stars indicate vesicular structures that are labelled by gold particles in the other image" is improper. The stars in two images obviously indicate different structures- the upper image indicates electron-dense structure and the lower image indicates peroxisomes (organelle). It is better to use different labels.

We thank the reviewers for coming up with “electron dense structures” term. This is a much better definition for the localization that we are seeing. We have made the changes in the legends and text as suggested by the reviewer. Here is the main text part where we mention electron microscopy data: Immunogold labelling in transmission electron micrographs of cells expressing GFP:PexRD54 revealed a strong signal in electron dense structures that are not peroxisomes (Figure 3—figure supplement 2–Figure 3—figure supplement 3, Figure 3—source data 1).

Here is the new legend of Figure 3—figure supplement 3:

“Figure 3—figure supplement 3. PexRD54 labelled high electron dense structures that are not peroxisomes. Serial sections of *N. benthamiana* leaves transiently expressing GFP:PexRD54 were collected three days post infiltration and probed with Anti-GFP and Anti-Catalase antibodies conjugated to gold particles. High electron dense structures labelled with GFP antibody were different than the regions labelled with catalase antibody, confirming that PexRD54 labelled structures are not peroxisomes. Stars indicate regions that are labelled by gold particles in the other image. Scale bar=500nm”.

For anti-GFP, the upper image indicates that gold particles were labeled in the electron-dense portion, which was neither associated with vesicle-like structures nor peroxisomes.

We thank the reviewers. We now define this localization pattern as electron dense structures.

2) About Figure 3—figure supplement 3:

Based on the total set of supplemental EM images which authors provided, Figure 3—figure supplement 2 seems not to be a typical image. Since most of images do not have vesicle like structures, the photo of number 11 seems to represent the total set of images.

We thank the reviewers for carefully investigating our images. We have replaced the old image with Image 11 from the raw data set.

3) Methods – Electron Microscopy and Immunogold labelling:

Since authors used high pressure freezing to get immunogold labelling, the authors need to describe the process in the Methods. I think authors may have cited the wrong reference (Liu, Schiff and Dinesh-Kumar et al., 2002), please check it.

In your revision, please include your archive of EM images as supplemental material so that readers can see the whole range of data.

We are already presenting all the images as a supplemental data set (Figure 3—source data 1). It will be available to the readers.